# Parts of Speech–Grounded Subspaces in Vision-Language Models

**James Oldfield**[1]* **Christos Tzelepis**[1] **Yannis Panagakis**[2,3]
**Mihalis A. Nicolaou**[4] **Ioannis Patras**[1]
[1]Queen Mary University of London   [2]National and Kapodistrian University of Athens
[3]Archimedes/Athena RC   [4]The Cyprus Institute

## Abstract

Latent image representations arising from vision-language models have proved immensely useful for a variety of downstream tasks. However, their utility is limited by their entanglement with respect to different visual attributes. For instance, recent work has shown that CLIP image representations are often biased towards specific visual properties (such as *objects* or *actions*) in an unpredictable manner. In this paper, we propose to separate representations of the different visual modalities in CLIP's joint vision-language space by leveraging the association between *parts of speech* and specific visual modes of variation (e.g. nouns relate to objects, adjectives describe appearance). This is achieved by formulating an appropriate component analysis model that learns subspaces capturing variability corresponding to a specific part of speech, while jointly minimising variability to the rest. Such a subspace yields disentangled representations of the different visual properties of an image or text in closed form while respecting the underlying geometry of the manifold on which the representations lie. What's more, we show the proposed model additionally facilitates learning subspaces corresponding to *specific* visual appearances (e.g. artists' painting styles), which enables the selective removal of entire visual themes from CLIP-based text-to-image synthesis. We validate the model both qualitatively, by visualising the subspace projections with a text-to-image model and by preventing the imitation of artists' styles, and quantitatively, through class invariance metrics and improvements to baseline zero-shot classification.

## 1   Introduction

Many recent advances in machine learning have been driven by vision-language (VL) models' ability to learn powerful, generalisable image representations from natural language supervision [1, 2, 3]. The image features from VL models well-capture representations of many visual attributes as evidenced by the broad applicability they have found for use in downstream tasks. The image or text encoders of CLIP in particular [1] have been used for controllable image synthesis [4, 5, 6], image captioning [7, 8], and multiple other discriminative tasks [9, 10, 11]. However, modelling the many different visual modalities in a single vector representation is not without its drawbacks–recent work shows that CLIP's visual representations are often *entangled*. For example, Goh et al. [12] find that specific neurons fire in response to both images containing a visual concept and *images of text* relating to the same concept. This leaves CLIP open to vulnerabilities in the form of 'Typographic attacks'–writing another class name as text on the image can often cause CLIP to predict this class with a higher probability than that of the original image's true category. Other recent works show that CLIP's visual representations encode task-specific attributes (such as of the object or action depicted in an image) in

---

*Corresponding author: `j.a.oldfield@qmul.ac.uk`

37th Conference on Neural Information Processing Systems (NeurIPS 2023).

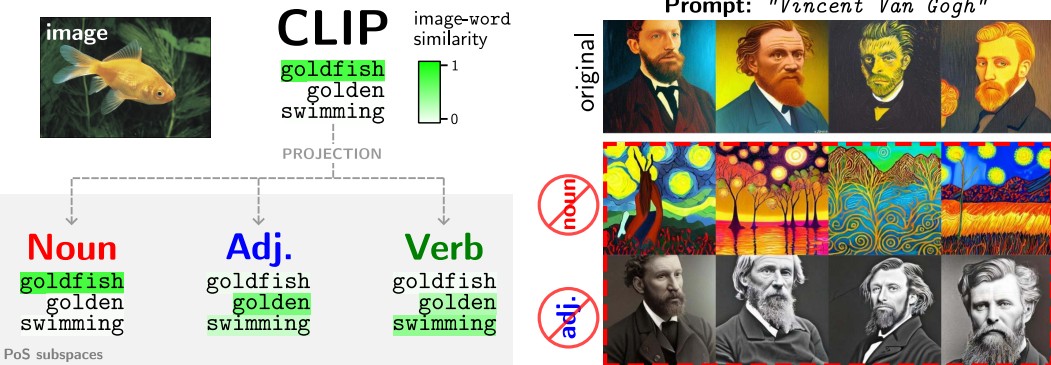

(a) Image-word similarity with both CLIP's embedding and after projecting it onto the PoS subspaces.

(b) Text-to-image visualisation of the subspace disentanglement of phrases with multiple visual associations.

Figure 1: CLIP represents multiple visual modes of variation in an embedding (e.g. the 'object' and its 'appearance'). The learnt PoS subspaces more reliably separate the constituent visual components.

an unpredictable manner, and often the embedding is biased in the prominence with which it encodes different modalities [13]. We show in Figure 1a a motivating example of this problem identified by Menon et al. [13]–the 'goldfish' noun embedding dominates the image's representation despite there being multiple additional labels which accurately describe important information about the image's contents. As a visual example, we find that CLIP encodings of text prompts containing 'visually polysemous' [14] phrases of artists' names lead CLIP-based text-to-image models [15] to synthesize an unpredictable combination of *both* images of the artists and of artworks in their signature styles (as shown in Figure 1b). Multiple visual associations of the text prompt, including both the appearance of the artist themselves and the style of their artwork, are entangled in the same CLIP embedding. For VL representations to make for useful image features, it's vital that the particular modalities of interest are indeed well-represented in the embedding. One popular means to this end is fine-tuning the representations for specific downstream tasks [9]. However, this not only requires additional computation but makes the restrictive assumption of the existence of labelled data for each task.

In this paper, we address the problem of better disentangling the modes of visual variation in CLIP's shared vision-language space. In particular, we ask the question: do there exist subspaces in CLIP's joint VL space that capture the representations of the 'content' of an image or text, that are invariant to its 'appearance'? To take the first steps towards achieving this we leverage the association between *parts of speech* in natural language and specific modes of visual variation: our learnt noun subspace isolates representations of the 'object' of an image or text prompt (e.g. an animal in an image, or the noun described in a sentence), and the adjective space its appearance (e.g. whether an object is shiny, or a scene is snowy). This image-text semantic similarity is instilled in the representations through CLIP's contrastive learning training objective, through which it is encouraged to learn image encodings that have large cosine similarity in the shared vision-language space with text encodings of captions describing its visual content. We show how by using example words in the parts of speech categories in the WordNet [16] database, one can extract representations of *both* image and text embeddings that better isolate the individual modes of visual variation. This is in contrast to the method of VL model 'prompting' [17, 18, 13], which often steers the embeddings of either only the text or image modality. We take inspiration from the related line of work that finds so-called 'interpretable directions' in latent space [19, 20, 21, 22, 23] that capture high-level semantic attributes–however, we learn subspaces that capture variation *uniquely* present amongst representations of words of particular parts of speech. We achieve this by formulating an appropriate objective function which we show can be manipulated into a well-known trace maximisation problem with a fast solution given in closed form. Since the CLIP representations live on the hypersphere [24], we further propose a manifold generalisation of the subspaces [25, 26] (illustrated in Figure 2) that share the property of capturing the variance of only the desired visual attributes, yet better respect the manifold on which the data lie. Concretely, we compute the proposed component analysis in the tangent space to the sphere's intrinsic mean, which can be seen as a local approximation of the manifold [27].

The method's ability to disentangle visual modes of variation is measured both qualitatively and quantitatively. Using a popular CLIP-based text-to-image model we demonstrate *visually* how the learnt PoS subspaces can better separate the content from the style associated with a text prompt. We show, for example, how simply projecting onto the orthogonal complements of the noun and adjective subspaces respectively can more reliably produce images of either the artists' work, or of the artists themselves, as shown in rows 2 and 3 of Figure 1b. We find removing a CLIP representations' adjective space component to be remarkably effective for preventing the visual imitation of artists' styles in this new class of CLIP-based text-to-image models [15], addressing societal concerns of the technology. Further, we show our objective additionally facilitates learning subspaces corresponding to more specific visual appearances (e.g. 'gory'). Projections onto the orthogonal complements of such subspaces consequently remove entire visual themes from text-to-image models, whilst preserving existing concepts through the PoS guidance.

Finally, we validate the subspaces' ability to isolate visual modes of variation quantitatively through a measure of class invariance (comparing to two existing baselines), and by showing how the baseline zero-shot classification protocol *in the learnt subspaces* leads to higher accuracies on the `ViT-B-32` CLIP model on $14/15$ datasets considered. Our contributions can be summarised as follows:

- We present a method for learning geometry-aware subspaces in CLIP's shared vision-language space that disentangle representations of the content in an image or text prompt from the way it looks, using parts of speech as supervision. To the best of our knowledge, we are the first to use the semantic relationship between parts of speech and specific visual modes of variation to disentangle CLIP's shared vision-language space.

- We formulate and solve an appropriate trace maximisation problem that admits a fast closed-form solution respecting the manifold on which the data lie.

- We validate our method's success at disentangling the visual modes of variation both quantitatively and qualitatively: by visually separating a text prompt's 'content' from its 'appearance' with text-to-image models for the former and through improved zero-shot classification in the submanifolds for the latter.

- We show the model's ability to further learn subspaces of more specific appearance-based variation (e.g. artists' styles), providing a way of erasing entire visual themes from CLIP-based text-to-image models.

## 2 Method

We first recall the basics of CLIP. We then motivate and introduce our objective function for learning parts of speech–specific linear subspaces in Section 2.1, and detail its closed-form solution in Section 2.1.1. Finally, in Section 2.2 we show how to learn *subspaces of the tangent space* to the CLIP VL hypersphere's intrinsic mean, which better respects the geometry of the manifold on which the CLIP representations lie.

**CLIP preliminaries**  Pre-trained image and text encoders map images and text respectively to latent representations $\mathbf{z}_I, \mathbf{z}_T \in \mathbb{R}^d$ in a shared 'vision-language' (VL) space [1]. For test-time zero-shot classification, candidate text prompts are first encoded with the text encoder. Then, the cosine similarity between an encoded image representation of interest and the text candidates is computed with $S(\mathbf{z}_T, \mathbf{z}_I) = \mathbf{z}_T^\top \mathbf{z}_I / (||\mathbf{z}_T|| \cdot ||\mathbf{z}_I||)$, after which the softmax function is applied to determine the most likely text label for the image. We now address the task of extracting representations of solely the target modes of variation in the section that follows.

### 2.1 Objective

We seek a lower-dimensional subspace on which to project either a text or image CLIP representation $\mathbf{z} \in \mathbb{R}^d$ to predictably isolate the desired visual mode of variation. We achieve this through natural language supervision in the form of words from the different parts of speech. Let the elements of a set $\mathcal{C}$ index into the relevant word class of interest (i.e. $\mathcal{C} = \{N, A, V, R\}$ for nouns, adjectives, verbs, and adverbs respectively), and $\mathbf{X}_i \in \mathbb{R}^{d \times n}, \forall i \in \mathcal{C}$ contain in their columns the CLIP encodings of $n$ words belonging to word class $i$ [16]. Then, for a word class $i \in \mathcal{C}$ of interest, we seek a $k$-dimensional subspace of $\mathbb{R}^d$ spanned by the columns of a learnt $\mathbf{W}_i \in \mathbb{R}^{d \times k}$ in which the CLIP

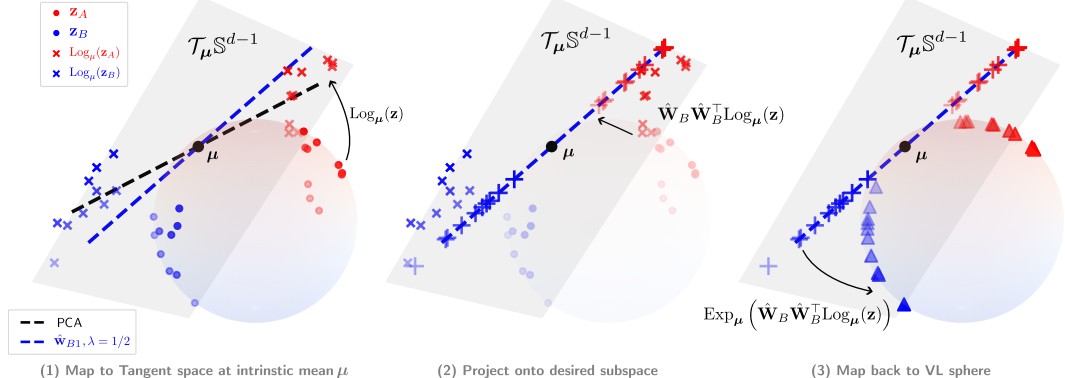

Figure 2: Our proposed method, shown on toy data, for learning 'geodesic submanifolds' [25] that capture the variance of visual attributes uniquely associated with specific parts of speech. After mapping to the tangent space (1), a linear subspace of the tangent space is learnt that captures the variation in only the target class $i$ (leading eigenvectors of $\hat{\mathbf{C}}_i$). Test samples can then be projected onto this subspace (2), and mapped back (3) to the VL sphere.

representations of the words in the class of interest $i$ have a large norm and the remaining categories' representations in $\mathcal{C} \setminus \{i\}$ are close to the zero vector. Intuitively, a hyperplane with this property models factors of variation that are uniquely present in representations of text that belong to a particular part of speech. We quantify this by formulating the following objective function:

$$\mathbf{W}_i = \underset{\mathbf{W}_i^\top \mathbf{W}_i = \mathbf{I}_k}{\arg\max} \left\{ (1-\lambda)||\mathbf{W}_i^\top \mathbf{X}_i||_F^2 - \sum_{j \in \mathcal{C} \setminus \{i\}} \lambda ||\mathbf{W}_i^\top \mathbf{X}_j||_F^2 \right\}, \tag{1}$$

where $\lambda \in [0, 1]$ is a hyperparameter that controls the importance of killing the variation in the non-target categories relative to preserving the variation in the target class.

### 2.1.1 Closed-form solution

Imposing column orthonormality on $\mathbf{W}_i$ not only ensures its columns span a full $k$-dimensional subspace, but also allows Equation (1) to be solved in closed form. Concretely, we first manipulate the objective as follows

$$(1-\lambda)||\mathbf{W}_i^\top \mathbf{X}_i||_F^2 - \sum_{j \in \mathcal{C} \setminus \{i\}} \lambda ||\mathbf{W}_i^\top \mathbf{X}_j||_F^2$$

$$= (1-\lambda)\mathrm{tr}\left( (\mathbf{W}_i^\top \mathbf{X}_i)^\top (\mathbf{W}_i^\top \mathbf{X}_i) \right) - \sum_{j \in \mathcal{C} \setminus \{i\}} \lambda \mathrm{tr}\left( (\mathbf{W}_i^\top \mathbf{X}_j)^\top (\mathbf{W}_i^\top \mathbf{X}_j) \right) \tag{2}$$

$$= \mathrm{tr}\left( \mathbf{W}_i^\top \left( (1-\lambda)\mathbf{X}_i \mathbf{X}_i^\top - \sum_{j \in \mathcal{C} \setminus \{i\}} \lambda \mathbf{X}_j \mathbf{X}_j^\top \right) \mathbf{W}_i \right) \tag{3}$$

$$= \mathrm{tr}\left( \mathbf{W}_i^\top \mathbf{C}_i \mathbf{W}_i \right), \tag{4}$$

where $\mathbf{C}_i = \left( (1-\lambda)\mathbf{X}_i \mathbf{X}_i^\top - \sum_{j \in \mathcal{C} \setminus \{i\}} \lambda \mathbf{X}_j \mathbf{X}_j^\top \right)$. Having now reformulated our original objective in Equation (1) as a (constrained) trace maximisation problem, its solution $\mathbf{W}_i$ is given in closed form[2] as the leading $k$ eigenvectors of $\mathbf{C}_i$.

For ease of presentation, we have assumed the number of data points in each class to be equal. One can account for class imbalance straightforwardly whilst retaining a solution in closed form however: multiplying each Frobenius norm term in the original objective of Equation (1) by $\frac{1}{n_p}$ (where $n_p$ is the number of columns of $\mathbf{X}_p, \forall p \in \mathcal{C}$) leads to $\mathbf{C}_i = \left( \frac{1-\lambda}{n_i} \mathbf{X}_i \mathbf{X}_i^\top - \sum_{j \in \mathcal{C} \setminus \{i\}} \frac{\lambda}{n_j} \mathbf{X}_j \mathbf{X}_j^\top \right)$ in Equation (4).

---

[2]*Corollary 4.3.39* of Horn and Johnson [28]. Note that all summands in $\mathbf{C}_i$ are symmetric.

**Special cases** To contrast the proposed objective with related decompositions, it's instructive to consider the resulting subspaces for extreme values of $\lambda$ (for zero-mean data). In the limit case of $\lambda := 0$ for example, $\mathbf{W}_i$ is given by the top principal components of the data points for one particular class $i$. Conversely, a value of $\lambda := 1$ gives the *bottom* principal components of datapoints in all other classes $j \in \mathcal{C} \setminus \{i\}$. Thus, $\lambda$ can be seen as providing a trade-off between hyperplanes well-capturing the variance of the attributes in the target class and lying near-orthogonal to the points of the remaining classes. Finally, one recovers regular PCA when $\mathbf{C}_i := \sum_{j \in \mathcal{C}} \frac{1}{n_j} \mathbf{X}_j \mathbf{X}_j^\top$.

## 2.2 From subspaces to submanifolds

Through CLIP's choice of the cosine similarity as an objective function during training, the unit-norm VL representations live on the *hypersphere* $\mathbf{z}_I, \mathbf{z}_T \in \mathbb{S}^{d-1} \subset \mathbb{R}^d$ [24]. However, subspace learning in the ambient Euclidean space $\mathbb{R}^d$ does not respect the underlying geometry of the manifold on which the data lie. An orthogonal projection of a vector onto the learnt Euclidean subspaces is not guaranteed to result in a vector that remains on the sphere, even if all the input data do. Motivated by this, we extend the component analysis for linear subspaces developed in Equation (1) to *geodesic submanifolds* [25]. As demonstrated in Fletcher et al. [25] for PCA, an analogous approximate projection onto the geodesic submanifolds capturing the desired variances can be made by applying the exact same component analysis of Equation (1) in the tangent space $\mathcal{T}_{\boldsymbol{\mu}} \mathbb{S}^{d-1} \subset \mathbb{R}^d$ to the VL sphere data's intrinsic mean $\boldsymbol{\mu} \in \mathbb{S}^{d-1}$ instead. To this end, we use the so-called *Logarithmic Map* $\mathrm{Log}_{\mathbf{p}} : \mathbb{S}^{d-1} \to \mathcal{T}_{\mathbf{p}} \mathbb{S}^{d-1}$, which maps points on the sphere to the tangent space at a reference point $\mathbf{p} \in \mathbb{S}^{d-1}$ and its inverse, the *Exponential Map* $\mathrm{Exp}_{\mathbf{p}} : \mathcal{T}_{\mathbf{p}} \mathbb{S}^{d-1} \to \mathbb{S}^{d-1}$ to map points back onto the hypersphere (whose well-known definitions are provided in the supplementary material).

We can then compute the subspace *of the tangent space to the intrinsic mean* spanned by the columns of a $\hat{\mathbf{W}}_i \in \mathbb{R}^{d \times k}$ for word class $i$ as the leading $k$ eigenvectors of $\hat{\mathbf{C}}_i = \sum_n (1 - \lambda) \mathrm{Log}_{\boldsymbol{\mu}}(\mathbf{x}_{in}) \mathrm{Log}_{\boldsymbol{\mu}}(\mathbf{x}_{in})^\top - \sum_{j \in \mathcal{C} \setminus \{i\}} \lambda \mathrm{Log}_{\boldsymbol{\mu}}(\mathbf{x}_{jn}) \mathrm{Log}_{\boldsymbol{\mu}}(\mathbf{x}_{jn})^\top$. We have again assumed an equal number of class data points purely for ease of presentation. This whole process is visualised in Figure 2.

By projecting onto the learnt subspaces of $\mathcal{T}_{\boldsymbol{\mu}} \mathbb{S}^{d-1}$, one can better *isolate* or *remove* the visual attributes associated with part of speech $i$. To isolate or kill the attributes we compute the projection onto the column space (i) or orthogonal complements (ii) respectively with

$$\text{(i) } \Pi_i(\mathbf{z}) = \mathrm{Exp}_{\boldsymbol{\mu}}(\hat{\mathbf{W}}_i \hat{\mathbf{W}}_i^\top \mathrm{Log}_{\boldsymbol{\mu}}(\mathbf{z})), \qquad \text{(ii) } \Pi_i^\perp(\mathbf{z}) = \mathrm{Exp}_{\boldsymbol{\mu}}\big((\mathbf{I}_d - \hat{\mathbf{W}}_i \hat{\mathbf{W}}_i^\top) \mathrm{Log}_{\boldsymbol{\mu}}(\mathbf{z})\big). \quad (5)$$

Projection onto the Euclidean subspaces can be computed using the projection matrices $\mathbf{P}_i = \mathbf{W}_i \mathbf{W}_i^\top$ and $\mathbf{P}_i^\perp = \mathbf{I}_d - \mathbf{P}_i$. The way in which one can extract representations relating to the visual modes of variation for the parts of speech is explored in detail in Section 3.

# 3 Experiments

Here we present both qualitative (Section 3.1) and quantitative (Section 3.2) experiments to validate the model's ability to disentangle the object in a CLIP text or image representation from its appearance. We henceforth use 'subspace' to refer throughout to the manifold generalisation from Section 2.2.

**Implementation details** For all experiments, we use the following 4 parts of speech: *nouns*, *adjectives*, *verbs*, and *adverbs*. Our labelled data points (with which we compute the closed-form solution of Equation (4)) for these parts of speech are given by the WordNet [16] database. There are a total of 112219, 18021, 7295, and 3910 text-string data points from each of the categories respectively, after filtering out any word that appears in two or more parts of speech. We set $\lambda := 1/2$ for all experiments. For all quantitative results, we use the base CLIP `ViT-B-32` model. Please see the supplementary material for ablation studies on $\lambda$ and experiments on additional CLIP architectures.

## 3.1 Qualitative results

### 3.1.1 Visual disentanglement

Recent CLIP-based text-to-image models (TTIMs) learn a mapping from the CLIP embeddings to synthetic images depicting visually a text prompt [15, 29, 30]–a process that Menon et al. [13]

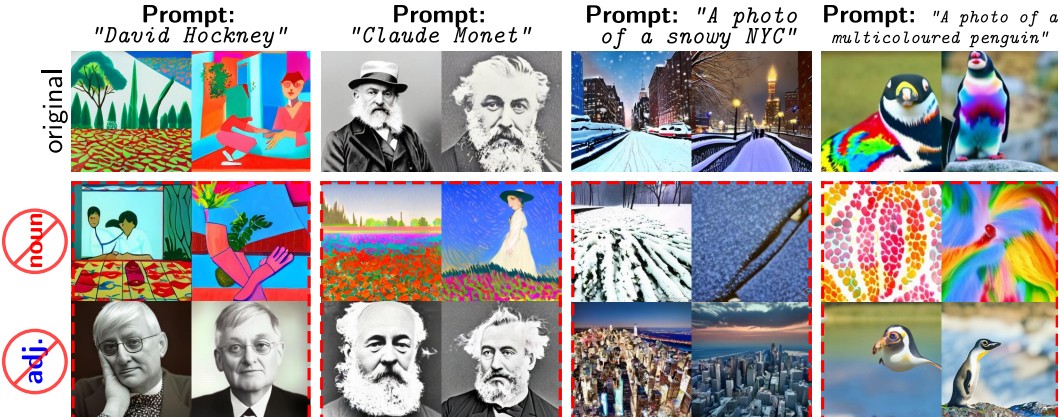

Figure 3: The synthetic images from the original CLIP representations (top row) and when first projecting them onto the orthogonal complements of the noun and adjective subspaces to remove the 'content' and 'appearance' component, respectively.

show is prone to also inheriting task bias. We begin in this section by following the experimental protocol of Materzynska et al. [31], using a recent popular CLIP-based TTIM [15] from LAION to demonstrate *visually* how our PoS subspaces can succeed in predictably isolating visual variation in the CLIP representations pertaining to either the object described in a text prompt or its appearance.

As one motivating example, we first study a curious instance of 'visually polysemous' [14] natural language phrases–terms that have multiple visual associations. In particular, we find that when prompted with artists' names, TTIMs produce *both* artworks in their style and images of the person themselves (e.g. the top row of Figure 3). That is to say, CLIP entangles in its representation these two dual meanings of the artists–their works' style and their physical appearance. We find that the PoS subspaces offer an intuitive way of reliably separating factors of visual variation in the representations for not just visually polysemous phrases, but also for natural language prompts more generally.

Concretely, projecting onto the orthogonal complement of the *noun* subspace $\Pi_N^\perp(\mathbf{z}_T)$ successfully removes from the embedding visual information about the object/content described implicitly in a text prompt $T$. For example, as visualised in row 2 of Figure 3 by reliably synthesising just artwork in an artist's style, or by leaving just the snowy or multicoloured appearance-based textures. On the other hand, projecting onto the adjective subspace's orthogonal complement $\Pi_A^\perp(\mathbf{z}_T)$ removes the visual appearances and styles associated with a text description (row 3 of Figure 3). For the visually polysemous artists' names, this isolates the representations of the artists themselves as humans–removing representations of their artwork styles. For the more complex prompts, this produces images of just the basic object described in the text prompt instead, such as the penguin or New York City. Many more examples of this visual disentanglement, details on the experimental setup, and ablation studies can be found in the supplementary material.

### 3.1.2 Style-blocking subspace projections

One societal concern with free-form TTIMs is their ability to produce imitation artworks copying the style of artists. Here, we show how the learnt adjective subspace can be used as is as a step towards mitigating this. To achieve this, one simply modifies the TTIM forward pass to first project the CLIP text representations onto the orthogonal complement of the adjective subspace with $\Pi_A^\perp(\mathbf{z}_T)$ before feeding it into the image generator. We see from the results in Figure 4 that this modification indeed prevents the imitation of the visual styles of a range of artists (even with multiple forms of sentence structure in the prompt), whilst still enabling a diverse set of images to be generated nonetheless.

**Visual theme subspaces** Whilst successfully preventing the visual imitation of many famous artists, applying the adjective subspace projection to *every* text prompt's CLIP representation can restrict the ability to use adjectives to specify visual appearance. We find a further effective strategy is to build additional subspaces for more specific visual appearances (e.g. artists' painting styles). Concretely,

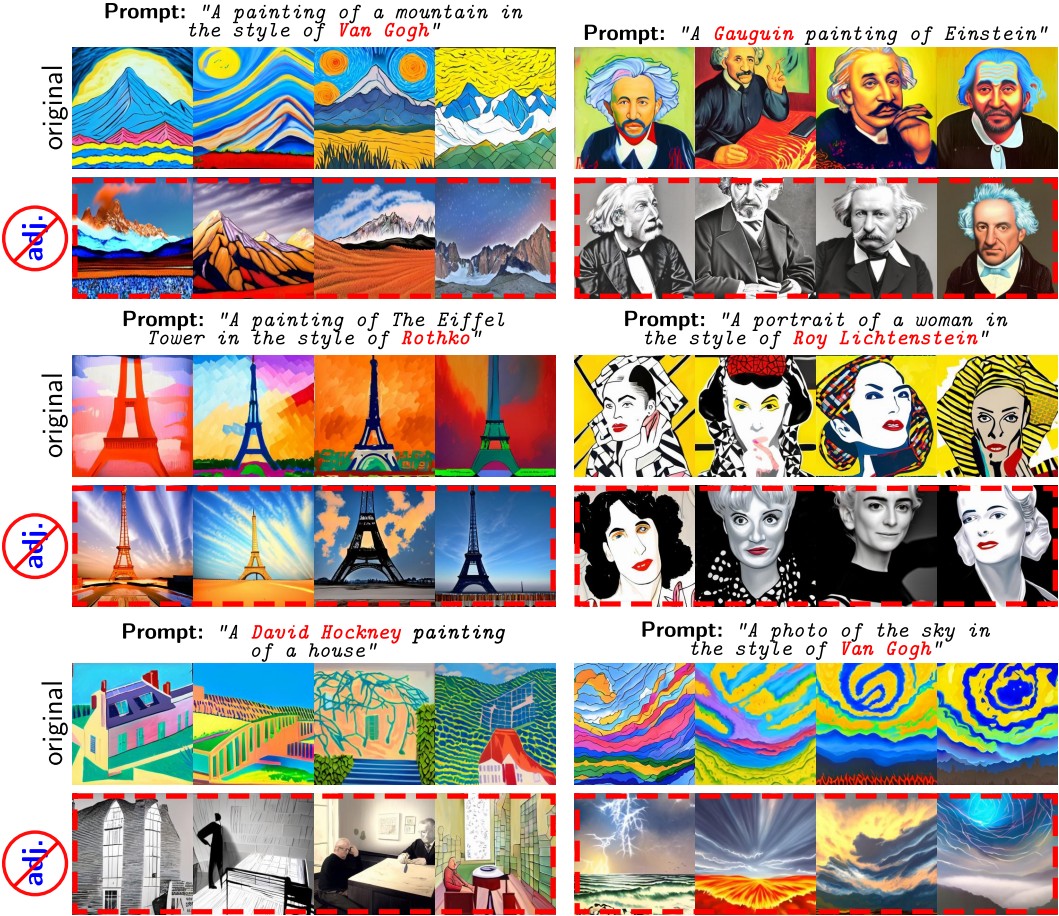

Figure 4: Killing the CLIP representations' component in the adjective subspace provides a way to block the imitation of artists' styles in CLIP-based text-to-image models.

we embed 830 artists' names and surnames in a new matrix $\mathbf{X}_{i'}$ and solve Equation (1) using all PoS classes in the negative summation to prevent the destruction of existing concepts. In contrast to the adjective space, projection onto the orthogonal complement of this 'artist subspace' *preserves* adjective-based visual descriptions whilst also successfully preventing style imitation (Figure 5a). Crucially, we highlight that for the example shown in Figure 5a, **the artist's name** `Qi Baishi` **is not present in the 'training' list of example artists**, suggesting the subspace has learnt a more general notion of an artist rather than simply the variation for only those artists whose names are provided as supervision.

We suggest more generally that subspaces learnt from a collection of words describing specific themes (whilst retaining variation in the PoS data through the main objective) could be useful for erasing entire visual concepts (such as NSFW imagery) from the CLIP representations. An additional example of another such custom subspace is shown in Figure 5b for removing only gory/bloody visual appearances. Not only does this offer a way to erase specific appearances in CLIP-based text-to-image models, but might also offer application in discriminative tasks, such as being able to block CLIP-based retrieval of images of sensitive nature. Please see the supplementary material for additional details, results, and visualisation of failure cases.

## 3.2 Quantitative results

Here we show quantitatively how the PoS–grounded subspaces can isolate the variation in the CLIP representation of either an image *or* text prompt. We validate this in two ways: through a class invariance metric in Section 3.2.1 and through zero-shot classification in Section 3.2.2.

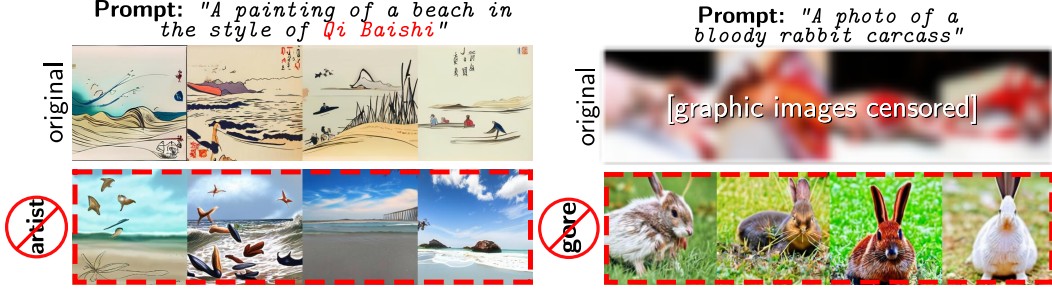

(a) A custom subspace for 'artistic style'.  (b) A custom subspace for gory/bloody visual themes.

Figure 5: Projecting onto the orthogonal complement of custom visual theme subspaces erases *specific* appearances from CLIP-based text-to-image models' images.

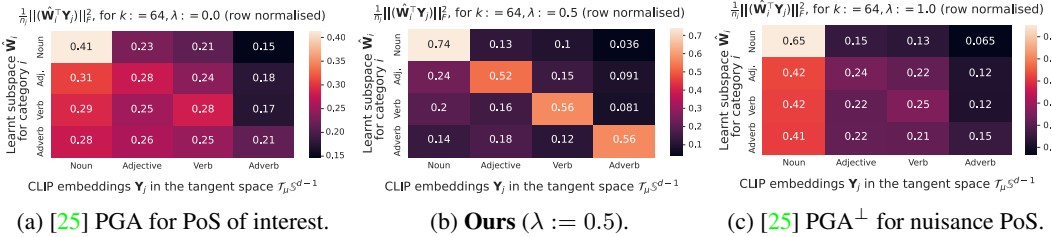

(a) [25] PGA for PoS of interest.  (b) **Ours** ($\lambda := 0.5$).  (c) [25] PGA$^\perp$ for nuisance PoS.

Figure 6: Subspace class invariance: The norm of the `ViT-B-32` CLIP text representations of 5k words from each PoS, in each of the subspaces. Results on baseline methods are shown in (a) and (c).

### 3.2.1 Class invariance

We first aim to quantify the extent to which our method results in subspaces that capture variation solely in the desired part of speech. We quantify this by a class invariance metric to measure how much each class' data points are retained in the subspaces. Concretely, we compute the Frobenius norm $\frac{1}{n_j}||\hat{\mathbf{W}}_i^\top \mathbf{Y}_j||_F^2$ where each column $\mathbf{y}_{jn} = \mathrm{Log}_{\boldsymbol{\mu}}(\mathbf{x}_{jn})$ contains the CLIP embeddings in the columns of $\mathbf{X}_j$ mapped to the tangent space, and $n_j$ is the number of data points in class $j$.

We compare the proposed method to two other component analyses performed in the tangent spaces. In Figure 6a we project onto a subspace learnt from only the data of the specific PoS of interest (e.g. the first row's subspace learnt through principal geodesic analysis [26]–maximising the variance for just the noun PoS data points), whilst in Figure 6c the subspace is learnt by simply minimizing the variance in the projections for the nuisance PoS classes (e.g. the first row's subspace minimizing the variance for the remaining adjective, verb, and adverbs' data points). We show the results of the proposed method with $\lambda = 0.5$ in Figure 6b for all combinations of parts of speech.

This quantity should be as close as possible to 0 in each of Figure 6's subfigures' off-diagonals $i \neq j$ and as close as possible to 1 along the figures' diagonals $i = j$ (if the subspaces capture visual variation that is unique to a particular word class). The learnt subspaces in Figure 6b evidently exhibit this pattern of disentanglement better than the baselines; the vectors have a large norm in their labelled class and a much smaller norm in the subspaces of the other categories.

### 3.2.2 Zero-shot subspace classification

In this subsection, our goal is to further validate quantitatively how well our noun subspace can isolate the representations relating to the content in an image by studying the task of zero-shot image

Table 1: Top-1 ZS classification accuracy with CLIP `ViT-B-32`.

| | ImageNET | MIT-states | UT Zap. | DomainNET | StanfordCars | Caltech101 | Food101 | CIFAR10 | CIFAR100 | OxfordPets | Flowers102 | Caltech256 | STL10 | MNIST | FER2013 |
|---|---|---|---|---|---|---|---|---|---|---|---|---|---|---|---|
| [1] CLIP | 56.50 | 45.30 | 82.20 | 50.60 | 58.50 | 80.00 | 75.20 | 87.50 | 60.40 | 73.80 | **62.50** | 80.10 | 96.10 | 29.10 | 37.80 |
| [32] PoS PCA | 56.50 | 45.40 | 81.80 | 50.60 | 58.30 | 80.10 | 75.20 | 87.50 | 60.40 | 73.80 | **62.50** | 80.10 | 96.10 | 29.10 | 37.60 |
| [25] PoS PGA | 56.70 | 45.20 | 84.40 | 50.40 | 56.20 | 80.00 | 75.20 | 86.10 | 60.20 | 75.10 | 60.20 | 80.50 | 95.80 | 29.40 | 38.00 |
| **Noun Submanifold** | **57.80** | **45.50** | **87.70** | **50.80** | **58.60** | **81.90** | **76.30** | **87.90** | **61.40** | **75.20** | 61.00 | **81.50** | **96.30** | **29.80** | **44.70** |

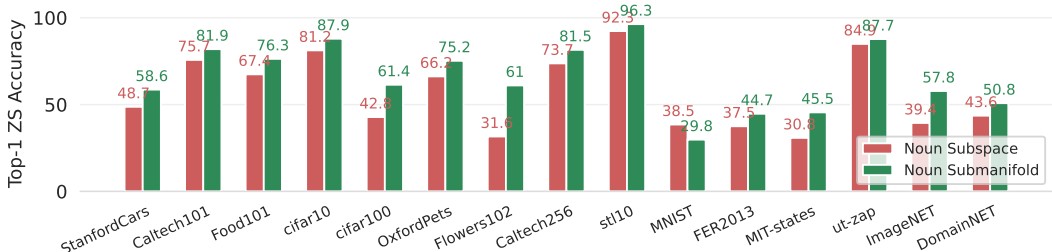

Figure 7: Ablation study on the zero-shot accuracy for the subspaces vs the submanifolds with CLIP `ViT-B-32`; the submanifold outperforms the subspaces on almost all datasets considered.

classification[3], given this is a very common application of CLIP in practice [1]. Importantly, these experiments serve as an additional way of measuring the subspaces' ability to isolate task-relevant modes of variation in the CLIP representations. With this in mind, we consider the baseline zero-shot setting [1] which computes the cosine similarity between all images in the training sets' and names of the class labels' CLIP representations. In our case, rather than computing the cosine similarity of the CLIP representations with $S(\mathbf{z}_I, \mathbf{z}_T)$ we measure instead the similarity in the *noun* subspace with $S\big(\Pi_N(\mathbf{z}_I), \mathbf{z}_T\big)$. We hypothesise that if the noun subspace indeed better isolates the object of an image–invariant to the particular style of the image–we should expect to see improvement to the baseline image classification.

We show the Top-1 accuracies for a large variety of datasets in Table 1. We find the noun submanifold projection to lead to improved zero-shot classification on $14/15$ of the datasets considered with CLIP `ViT-B-32`, and include a comparison to the Euclidean subspaces in Figure 7 to further illustrate the benefit of the geometry-aware variant. This is without needing any prompt engineering or domain knowledge about each dataset separately, confirming that the subspaces serve the intended role of being able to isolate the visual modalities of interest automatically. For all results in this subsection, we project onto a relatively large $k := 500$ dimensional subspace for PCA, PGA, and the proposed method. Please see the supplementary material for results on 2 more alternative CLIP architectures.

## 4   Related work

**Vision-language representations**   There has been much interest in studying the properties of the VL representations in large-scale models such as CLIP. In particular, Goh et al. [12] show that there exist neurons that fire in response to both visual representations of a concept as well as to text relating to the concept. Viewing this as a form of *entanglement*, Materzynska et al. [31] address this by learning a linear subspace in which written text is disentangled from its visual component. Whilst we also wish to disentangle visual concepts, we do not focus on disentangling written words from visual representations, but rather visual modalities more generally. Further, our solution is instead given in closed form and has far fewer hyperparameters to tune.  In addition to disentangling written/visual concepts, a number of works explore other forms of disentanglement in multimodal models. For example, to better perform image captioning [35, 36], generation [37, 38, 39, 40, 41], or discovering compositional representations or structure [37, 38, 39, 42].

Another popular approach to learning more useful VL representations is fine-tuning; Ilharco et al. [43] use fine-tuning in combination with weight interpolation to improve results on specific tasks without affecting existing performance, whilst many other works fine-tune CLIP for specific domains such as action recognition [11], video recognition [10], image captioning [7], and more [9]. Despite the success of fine-tuning, this comes with the restrictive requirement of task-specific labels. The method of 'prompting' [44, 17, 18, 13] is one alternative approach. At a high level, this technique learns a set of parameters (either in the form of 'words' in a prompt [17, 44] or as pixels appended to an input image [18, 13]) to steer the input to produce more useful representations for specific downstream tasks. In contrast, our method disentangles the modes of variation directly in the joint

---

[3]For example, the colour in which a digit is drawn [33, 34] is not relevant information for the task of classifying which number it is.

vision-language representation space–this facilitates the ability to separate semantic information in *both* image and text representations. Consequently, the proposed method has direct application for both discriminative tasks and CLIP text-driven generative tasks.

**Component analysis**   Let the matrices $\mathbf{X}_1, \mathbf{X}_2 \in \mathbb{R}^{d \times n}$ contain $n$ data points of two classes in their columns. The *Principal Component Analysis* (PCA) [32, 45] computes the low-dimensional subspace of maximum variance as the leading eigenvectors of the empirical covariance matrix of all data points. One pertinent drawback of PCA is its unsupervised nature in disregarding the labels of the data points. Whilst one can learn class-specific subspaces via PCA on the relevant subset of data (i.e. the eigenvectors of zero-mean $\frac{1}{n}\mathbf{X}_1\mathbf{X}_1^\top$), there is nothing to prevent the remaining 'nuisance' class(es) from also having large variance in this learnt subspace. One method that provides a way to jointly maximise the variance in one class' projected embeddings whilst minimizing that of another class is the *Fukunaga-Koontz transform* [46, 47] (FKT). FKT learns a class 1–specific subspace spanned by the columns of $\mathbf{W}_1 \in \mathbb{R}^{d \times k}$ as the leading eigenvectors of $\left(\mathbf{U}\mathbf{D}^{-\frac{1}{2}}\right)^\top \mathbf{X}_1\mathbf{X}_1^\top\left(\mathbf{U}\mathbf{D}^{-\frac{1}{2}}\right)$, where $\mathbf{U}, \mathbf{D}$ are the eigenvectors (stacked column-wise) and eigenvalues (along the diagonal) respectively of the matrix $\left(\mathbf{X}_1\mathbf{X}_1^\top + \mathbf{X}_2\mathbf{X}_2^\top\right)$. Whilst FKT also learns a subspace in which solely the target class' points have a large coefficient, the FKT operates on just two classes of interest and is more computationally expensive than the proposed method in requiring two eigendecompositions (rather than one). Another related supervised decomposition is the *Fisher Discriminant Analysis* (FDA) [48] which learns a subspace in which classes can be easily discriminated. Despite FDA's objective involving a similar trace maximisation form to Equation (4), the goals of FDA and the proposed objective are very different. FDA aims to *minimise* intra-class variation, whilst the proposed objective *maximises* the norm of vectors of the target class in the learnt subspaces, whilst minimising those of the remaining classes. Additionally, the proposed objective does not suffer from the same restrictively small upper bound on the dimensionality of the learnt subspace as FDA does through the low-rank of the between-scatter matrix. A comparison to the subspaces learnt with FDA, FKT, and the proposed objective is shown in the supplementary material to illustrate the differences visually.

## 5   Conclusion

In this paper, we have proposed a method for learning geometry-aware subspaces in CLIP's vision-language space that disentangle the modes of visual variation in representations of both images and text. To achieve this, we used the semantic relationship between parts of speech in natural language and specific visual modes of variation. We demonstrated the disentanglement qualitatively with a text-to-image model, showcasing the model's ability to remove visual imitation of artists' styles from synthetic images. Class invariance metrics and zero-shot classification results further validated the disentanglement quantitatively.

**Limitations**   A common drawback of subspace learning approaches is choosing a good number of dimensions $k$. Our method inherits this limitation, and one must choose the appropriate value for the specific task. Despite this, the closed-form eigensolution means only a single fast computation is needed, and any desired number of eigenvectors can be used at test-time. Although the recovered subspaces show wide applicability in downstream tasks, they are not able to perfectly separate the modes of variation for every possible image and text prompt. Additionally, the PoS supervision can not help address polysemy *within* a particular part of speech (such as disambiguating between the bird/machine meanings of the word 'crane'). This further alludes to the promise of the custom subspaces in being able to target more specific visual concepts.  Finally, often even minor changes to text-to-image models' text input or its representation produces very different output images [49]. As can be seen from our results, the subspace projection of the entire text representation similarly does not guarantee the preservation of the structure of the images generated from the 'original' prompts.

## 6   Acknowledgements

This research was partially supported by the EU's Horizon 2020 programme H2020-951911 AI4Media project, a grant from The Cyprus Institute on Cyclone, and by project MIS 5154714 of the National Recovery and Resilience Plan Greece 2.0 funded by the European Union under the NextGenerationEU Program.

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

# A  Definitions and derivations

## A.1  Closed-form solution

Detailed steps for the expansion of the original objective into the trace maximisation form of the main objective are given as follows:

$$(1 - \lambda)||\mathbf{W}_i^\top \mathbf{X}_i||_F^2 - \sum_{j \in \mathcal{C} \setminus \{i\}} \lambda ||\mathbf{W}_i^\top \mathbf{X}_j||_F^2 \tag{6}$$

$$= (1 - \lambda)\mathrm{tr}\left((\mathbf{W}_i^\top \mathbf{X}_i)^\top (\mathbf{W}_i^\top \mathbf{X}_i)\right) - \sum_{j \in \mathcal{C} \setminus \{i\}} \lambda \mathrm{tr}\left((\mathbf{W}_i^\top \mathbf{X}_j)^\top (\mathbf{W}_i^\top \mathbf{X}_j)\right) \tag{7}$$

$$= \mathrm{tr}\left((1 - \lambda)\mathbf{X}_i^\top \mathbf{W}_i \mathbf{W}_i^\top \mathbf{X}_i - \sum_{j \in \mathcal{C} \setminus \{i\}} \lambda \mathbf{X}_j^\top \mathbf{W}_i \mathbf{W}_i^\top \mathbf{X}_j\right) \tag{8}$$

$$= \mathrm{tr}\left((1 - \lambda)\mathbf{W}_i^\top \mathbf{X}_i \mathbf{X}_i^\top \mathbf{W}_i - \sum_{j \in \mathcal{C} \setminus \{i\}} \lambda \mathbf{W}_i^\top \mathbf{X}_j \mathbf{X}_j^\top \mathbf{W}_i\right) \tag{9}$$

$$= \mathrm{tr}\left(\mathbf{W}_i^\top ((1 - \lambda)\mathbf{X}_i \mathbf{X}_i^\top - \sum_{j \in \mathcal{C} \setminus \{i\}} \lambda \mathbf{X}_j \mathbf{X}_j^\top)\mathbf{W}_i\right) \tag{10}$$

$$= \mathrm{tr}\left(\mathbf{W}_i^\top \mathbf{C}_i \mathbf{W}_i\right), \tag{11}$$

where $\mathbf{C}_i = \left((1 - \lambda)\mathbf{X}_i \mathbf{X}_i^\top - \sum_{j \in \mathcal{C} \setminus \{i\}} \lambda \mathbf{X}_j \mathbf{X}_j^\top\right)$. Here we have used $||\mathbf{X}||_F^2 = \mathrm{tr}\left(\mathbf{X}^\top \mathbf{X}\right)$ and the linearity and cyclic properties of the trace.

## A.2  The Logarithmic and Exponential Maps

For mapping to the hypersphere's tangent space at a reference point in the main paper, we use well-known explicit formulas. Concretely, the *Logarithmic Map* $\mathrm{Log}_{\mathbf{p}} : \mathbb{S}^{d-1} \to \mathcal{T}_{\mathbf{p}} \mathbb{S}^{d-1}$, which maps points on the sphere to the tangent space at a reference point $\mathbf{p} \in \mathbb{S}^{d-1}$ is defined as

$$\mathrm{Log}_{\mathbf{p}}(\mathbf{z}) = \arccos(\mathbf{z}^\top \mathbf{p}) \frac{(\mathbf{I}_d - \mathbf{p}\mathbf{p}^\top)(\mathbf{z} - \mathbf{p})}{||(\mathbf{I}_d - \mathbf{p}\mathbf{p}^\top)(\mathbf{z} - \mathbf{p})||_2}. \tag{12}$$

Its inverse, the *Exponential Map* $\mathrm{Exp}_{\mathbf{p}} : \mathcal{T}_{\mathbf{p}} \mathbb{S}^{d-1} \to \mathbb{S}^{d-1}$ mapping points back onto the sphere is given by

$$\mathrm{Exp}_{\mathbf{p}}(\mathbf{z}) = \cos\left(||\mathbf{z}||_2\right) \mathbf{p} + \sin\left(||\mathbf{z}||_2\right) \frac{\mathbf{z}}{||\mathbf{z}||_2}. \tag{13}$$

# B  Additional results

## B.1  Qualitative results

**Visual disentanglement**  Firstly, we include many more examples of the visual disentanglement with the 'adjective' and 'noun' PoS subspaces and TTIM of Rampas et al. [15] in Figures 8 and 9.

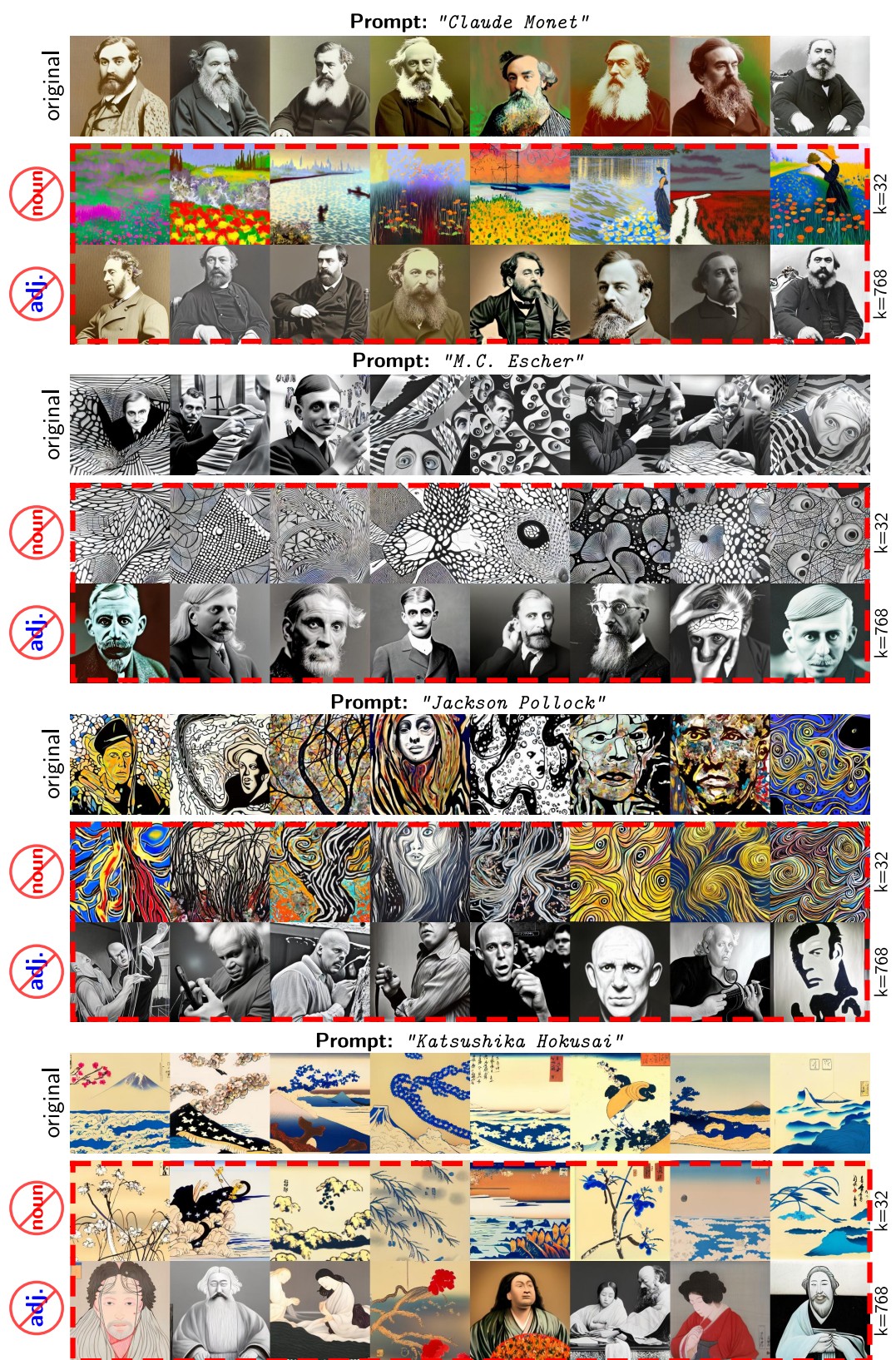

Figure 8: Additional results for visual disentanglement of text prompts using the learnt PoS subspaces.

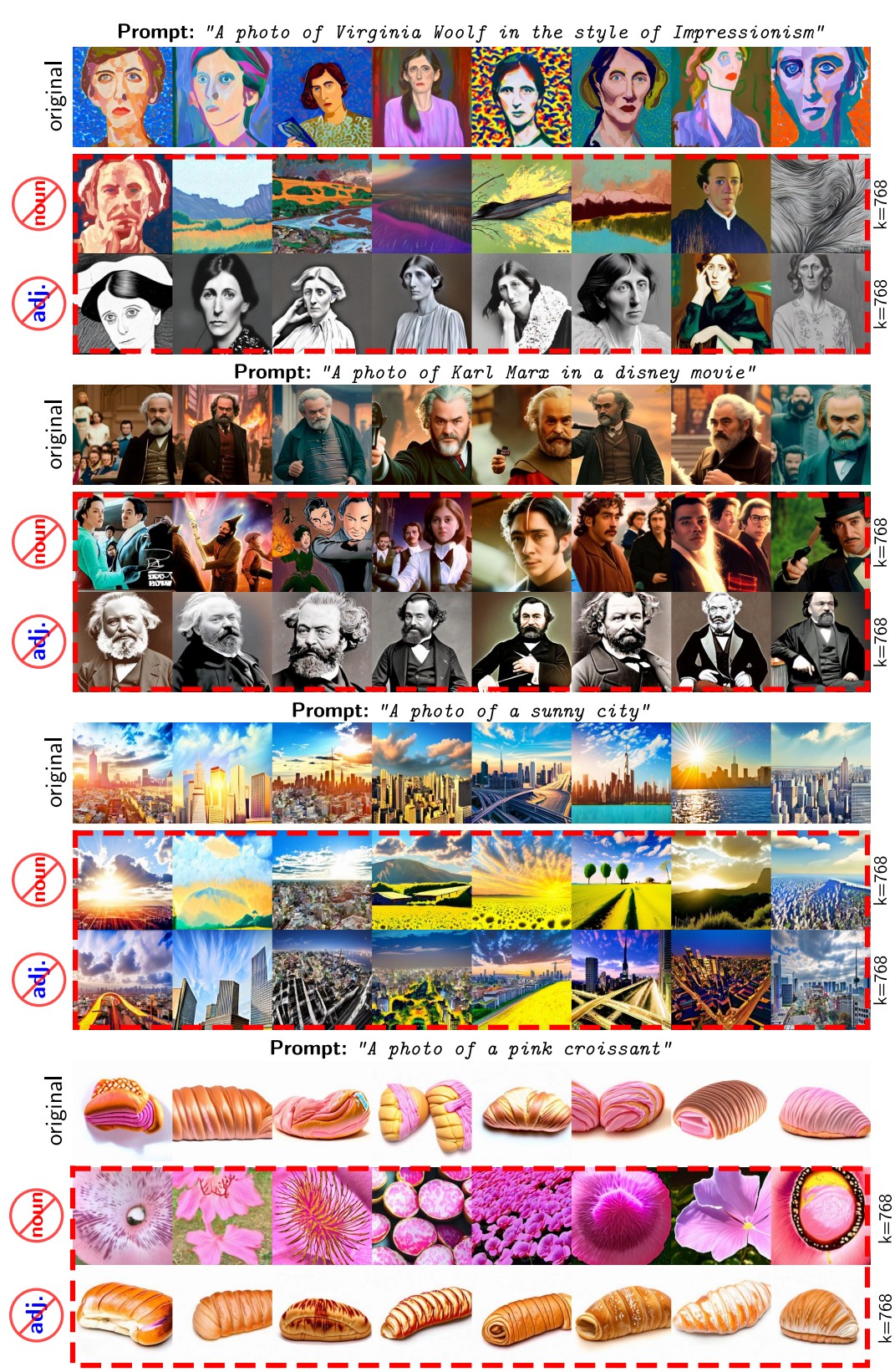

Figure 9: Additional results for visual disentanglement of text prompts using the learnt PoS subspaces.

**Visual theme removal**    As shown in the main paper, the adjective subspace works remarkably well for preventing the imitation of artists' styles in the CLIP-based text-to-image model Paella. However, as stated in the main paper, the adjective subspace orthogonal projection for the task of 'style blocking' is overly restrictive in also preventing the description of visual appearance with adjectives. We provide in Figure 10 some examples of this–for example, the 'stormy' and 'red' visual appearances are removed in Figure 10 after projection. On the other hand, the custom visual theme subspaces can target *specific* visual appearances more precisely–two examples are shown in Figure 11 (following the experimental setup outlined in Appendix D).

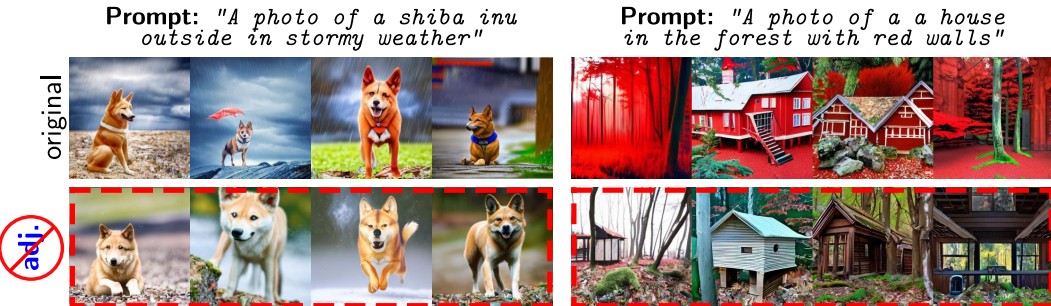

Figure 10: 'Style blocking' with the adjective subspace is overly restrictive (here blocking 'stormy' and 'red' visual appearances).

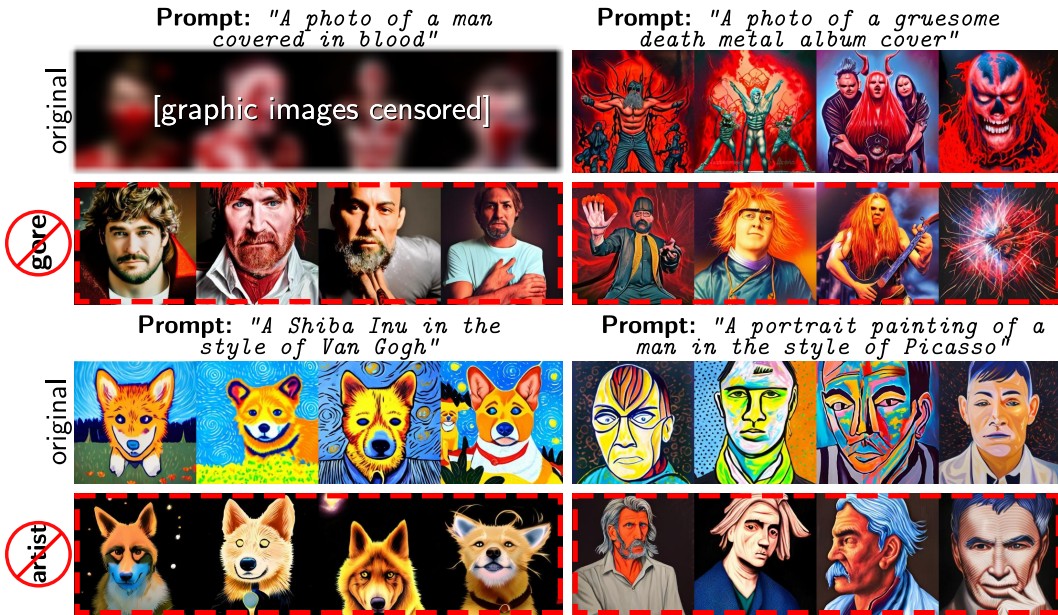

Figure 11: Additional results for the two custom visual theme orthogonal complement subspace projection for 'gore' and artists' styles.

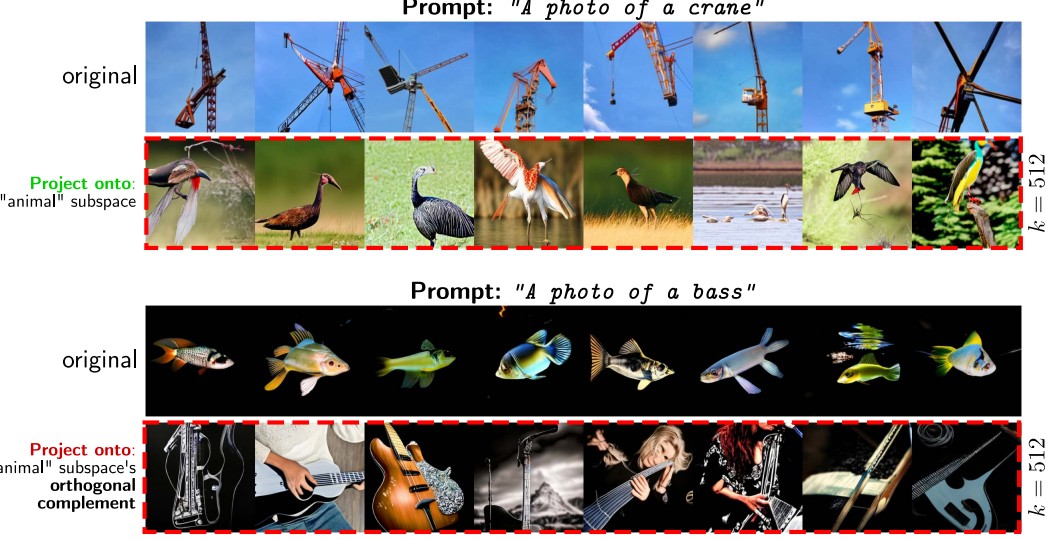

Figure 12: Projecting onto a custom subspace (given by 2758 names of animals) and its orthogonal complement learnt using the proposed objective. Such custom visual subspaces show promise in helping disambiguate between polysemous nouns.

**Polysemy within a single part-of-speech** As described in the limitations, the PoS subspaces cannot disambiguate between polysemous phrases *within* one particular PoS. We show in Figure 12 a proof-of-concept result suggesting that the custom visual subspaces can be a means of addressing this. In particular, we learn a custom "animal" subspace following the same protocol in Sect. 3.1.2 of the main paper. When projecting CLIP representations of "a photo of a crane" onto this subspace, we produce photos of the bird, rather than of machinery. Conversely, projecting the CLIP representation of "A photo of a bass" onto this subspace's orthogonal complement removes the representation of "bass" as a fish, synthesizing instead just images of bass guitars.

## B.2 Quantitative results

**Visual theme subspace invariance** Here we show quantitative results for the visual theme-specific subspaces. We have evaluated this visually via text-to-image models in Figure 11, however here we wish to demonstrate that large variation in the CLIP representations directly is captured for only the themes of interest. Concretely, in Figure 13 we compute the same class invariance metric for 200 random WordNET words and random theme-specific words in the learnt custom theme subspaces through $\frac{1}{200}||(\hat{\mathbf{W}}_i^\top \mathbf{Y})||_F^2$. Here, $\mathbf{Y} \in \mathbb{R}^{d \times 200}$ contains in its columns the CLIP word embedding mapped to the tangent space, and $i$ denotes the specific custom visual theme of interest for a particular subfigure. As can be seen from the high magnitude in the orange bars and low magnitude of the blue bars in Figure 13, these subspaces map the 200 random other words much closer to the zero vector than the theme-specific words. This indicates the variance in just the words of interest has indeed been captured.

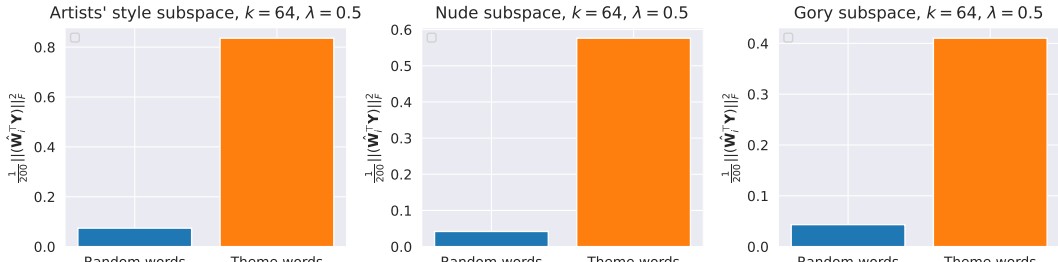

Figure 13: $\frac{1}{200}||(\hat{\mathbf{W}}_i^\top \mathbf{Y})||_F^2$ of CLIP representations (of both 200 random words and 200 words from the 'training' set describing the theme of interest) projected onto theme $i$-specific subspaces.

**Additional zero-shot classifcation** We show in both Table 2 and Table 3 additional results for the baseline zero-shot classification protocol (following the exact same setup in the main paper) with the similarity metric $S(\Pi_N(\mathbf{z}_I), \mathbf{z}_T)$ after the noun subspace projection. As can be seen, the proposed subspace leads to improved zero-shot classification on a wide range of datasets, for multiple CLIP architectures.

Table 2: Top-1 ZS classification accuracy with CLIP `ViT-B-16`.

| | ImageNET | MIT-states | UT Zap. | DomainNET | StanfordCars | Caltech101 | Food101 | CIFAR10 | CIFAR100 | OxfordPets | Flowers102 | Caltech256 | STL10 | MNIST | FER2013 |
|---|---|---|---|---|---|---|---|---|---|---|---|---|---|---|---|
| CLIP | 61.60 | 47.80 | 89.10 | 54.40 | 62.50 | 82.00 | 81.50 | 88.80 | 62.40 | 76.10 | 65.20 | 81.70 | 96.90 | 48.00 | 41.90 |
| PoS PCA | 61.60 | 47.90 | 89.00 | 54.40 | 62.50 | 82.10 | 81.50 | 88.80 | 62.30 | 76.00 | 65.30 | 81.80 | **97.00** | 47.70 | 41.70 |
| PoS PGA | 61.80 | 47.70 | **89.60** | 54.30 | 60.00 | 82.20 | 81.70 | 87.90 | 62.70 | **78.60** | 64.60 | 82.30 | 96.60 | 50.00 | 42.70 |
| **Noun Submanifold** | **62.80** | **48.30** | 88.10 | **54.70** | **62.70** | **82.90** | **82.10** | **89.10** | **63.90** | 77.00 | **65.40** | **83.40** | **97.00** | **55.00** | **48.60** |

Table 3: Top-1 ZS classification accuracy with CLIP `ViT-L-14`.

| | ImageNET | MIT-states | UT Zap. | DomainNET | StanfordCars | Caltech101 | Food101 | CIFAR10 | CIFAR100 | OxfordPets | Flowers102 | Caltech256 | STL10 | MNIST | FER2013 |
|---|---|---|---|---|---|---|---|---|---|---|---|---|---|---|---|
| CLIP | 69.00 | 51.80 | 90.40 | 59.60 | 74.70 | 80.90 | 87.40 | **91.70** | 72.70 | 82.50 | 74.60 | 86.80 | **97.70** | **59.70** | 35.70 |
| PoS PCA | 69.00 | 51.80 | 90.40 | 59.60 | 74.70 | 80.90 | 87.40 | **91.70** | 72.70 | 82.50 | 74.60 | 86.80 | **97.70** | **59.70** | 35.70 |
| PoS PGA | 69.10 | 51.90 | **90.50** | 59.60 | 74.60 | 80.80 | 87.50 | 91.60 | 72.80 | 82.60 | **74.80** | 86.80 | **97.70** | **59.70** | 35.80 |
| **Noun Submanifold** | **70.10** | **52.50** | 90.10 | **60.00** | **76.00** | **82.80** | **87.70** | 90.60 | **73.90** | **83.10** | 74.70 | **88.10** | 96.90 | 58.90 | **46.40** |

# C Ablation studies

## C.1 Relationship to alternative component analyses

We first compare 1D subspaces learnt with our method, FDA [48], and FKT [46] shown in Figure 14 on toy data chosen to illustrate the qualitative differences in the properties of the subspaces. Given the very different goals of FDA in minimising intra-class variation, the resulting FDA subspace is near-orthogonal to that of FKT and the proposed method. Whilst the FKT-given subspace is very close to ours, for this particular illustrative toy data our subspace better kills the variance in the red data (as illustrated in Figure 14 b.iii). This figure also provides a visualisation of the learnt subspace's proximity to the principal component of the target class and the bottom principal component of the red class' datapoints–the proximity to the two extremes being controlled by $\lambda$.

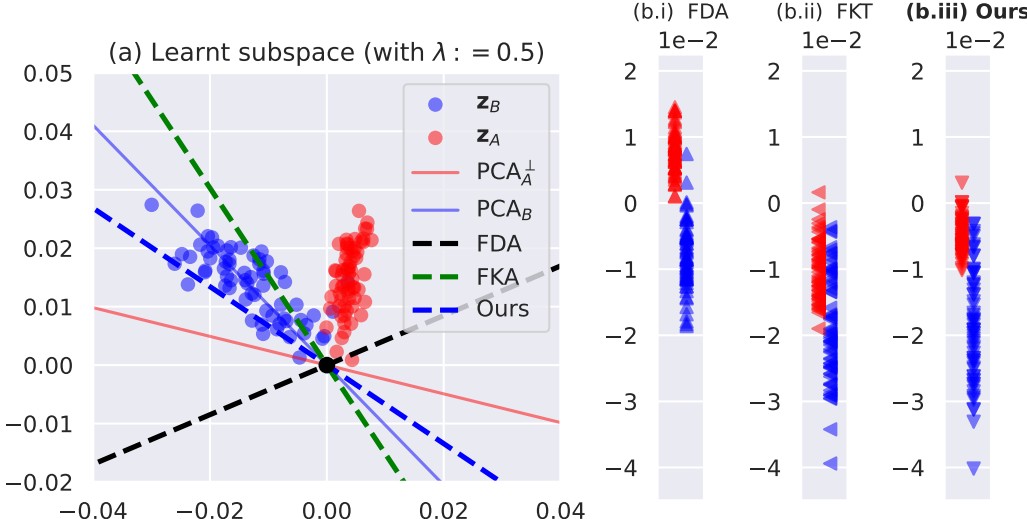

Figure 14: (a) A visual comparison of the leading eigenvector of $\mathbf{C}_i$ to the first FDA component (centred for comparison), to the first FKT component. Shown in (b) are the points' coordinates in the three subspaces. As can be seen, the learnt $\mathbf{w}_{B1}$ captures large variance in the blue target class and is close to orthogonal to data points of the red class.

## C.2 Role of $k$

We show the impact of various choices of $k$ (dimensionality of the subspaces) as visualised with text-to-image models in Figure 15 with the projections onto the orthogonal complements to kill the undesired variation. As can be seen, increasing $k$ removes more and more visual information relevant to the particular visual mode. For example, we see the image feature increasingly less snow on the right, whilst increasing less of London on the left.

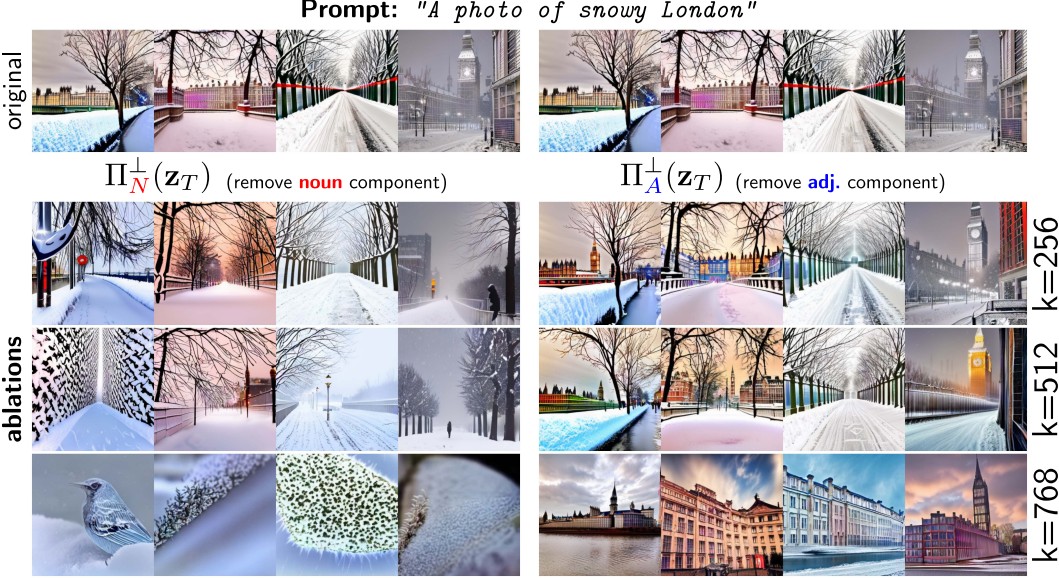

Figure 15: Ablation study on the value of $k$ on the $d = 1024$ OpenCLIP VL space when projecting onto the orthogonal complements of $k$-dimensional adjective and noun subspaces.

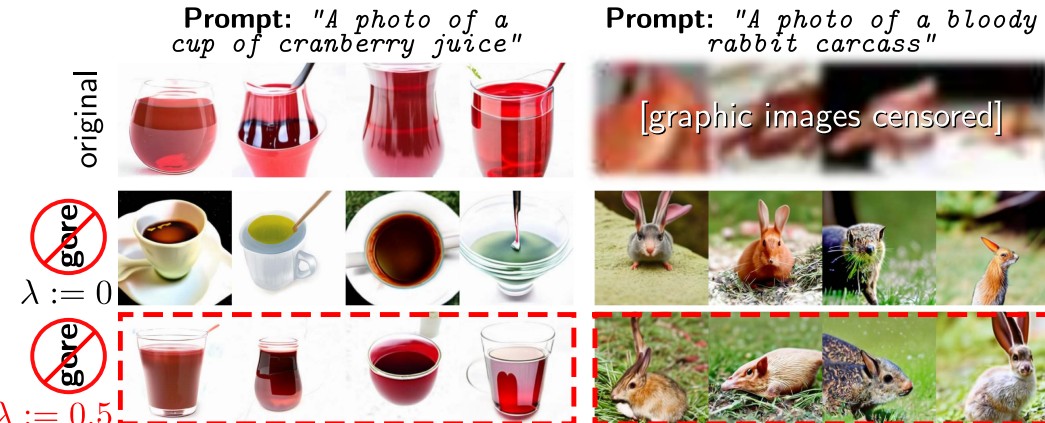

Figure 16: Without using PoS as 'negative guidance' (i.e. when $\lambda := 0$), related concepts (e.g. 'cranberry juice') can be visually removed to a much greater extent than when using the PoS guidance ($\lambda := 0.5$).

## C.3 Role of $\lambda$

We next provide an ablation study on the value of $\lambda$. We provide both a visual qualitative ablation and a quantitative one.

**Qualitative**  Concretely, in each subfigure (row) of Figure 17, we take the first 5000 WordNET text strings from each part of speech and compute their CLIP text embeddings $\mathbf{z}_T \in \mathbb{R}^d$. We then calculate $\hat{\mathbf{W}}_i^\top \text{Log}_{\boldsymbol{\mu}}(\mathbf{z}_T)$, plotting the first coordinate along the $x$-axis and the second along the $y$-axis of each subfigure in Figure 17. Ideally, the data points in the target class should be the only ones with a large norm if this hyperplane captures visual variation that is unique to a particular word class. We see that $\lambda := 0$ preserves the most variance in the target class' embeddings but the different categories' projections are clearly entangled–the other classes' datapoints also have large norm. Conversely, $\lambda := 1.0$ maps all points effectively to the zero vector–killing the variance in all categories. As can be seen in Figure 17c, $\lambda := 0.5$ offers a reasonable balance of both properties in this 4-class setting. The exact same experiments are run on the larger version of CLIP, shown in Figure 18, where similar conclusions can be drawn about the practical impact of $\lambda$.

**Quantitative**  For quantifying this in more dimensions, we compute the class invariance metric (used in the main paper) in Figure 19 and Figure 20 for various values of $\lambda$, where we observe that $\lambda := 0.5$ is a sensible choice for multiple CLIP architectures.

**Visual subspaces**  Finally, we demonstrate the importance of using PoS as 'negative examples' in the summation in the main objective when learning visual theme-specific subspaces. Intuitively, whilst we want to maximise the variation for phrases of a particular theme (such as 'gory'), we also want to preserve the ability to generate other concepts with the TIIM, which is what the objective provides through the hyperparameter $\lambda$.

In particular, we show in the second row of Figure 16 the visual results when projecting onto the orthogonal complement of a 'gory' subspace learnt when we do *not* use the PoS as 'negative guidance' (i.e. when $\lambda := 0$). As can be seen in comparison to the third row of Figure 16, using the PoS is critical in this instance for retaining the ability to synthesise existing related concepts (here ensuring e.g. 'cranberry juice' can still be synthesised even though visually 'gory' appearances are removed).

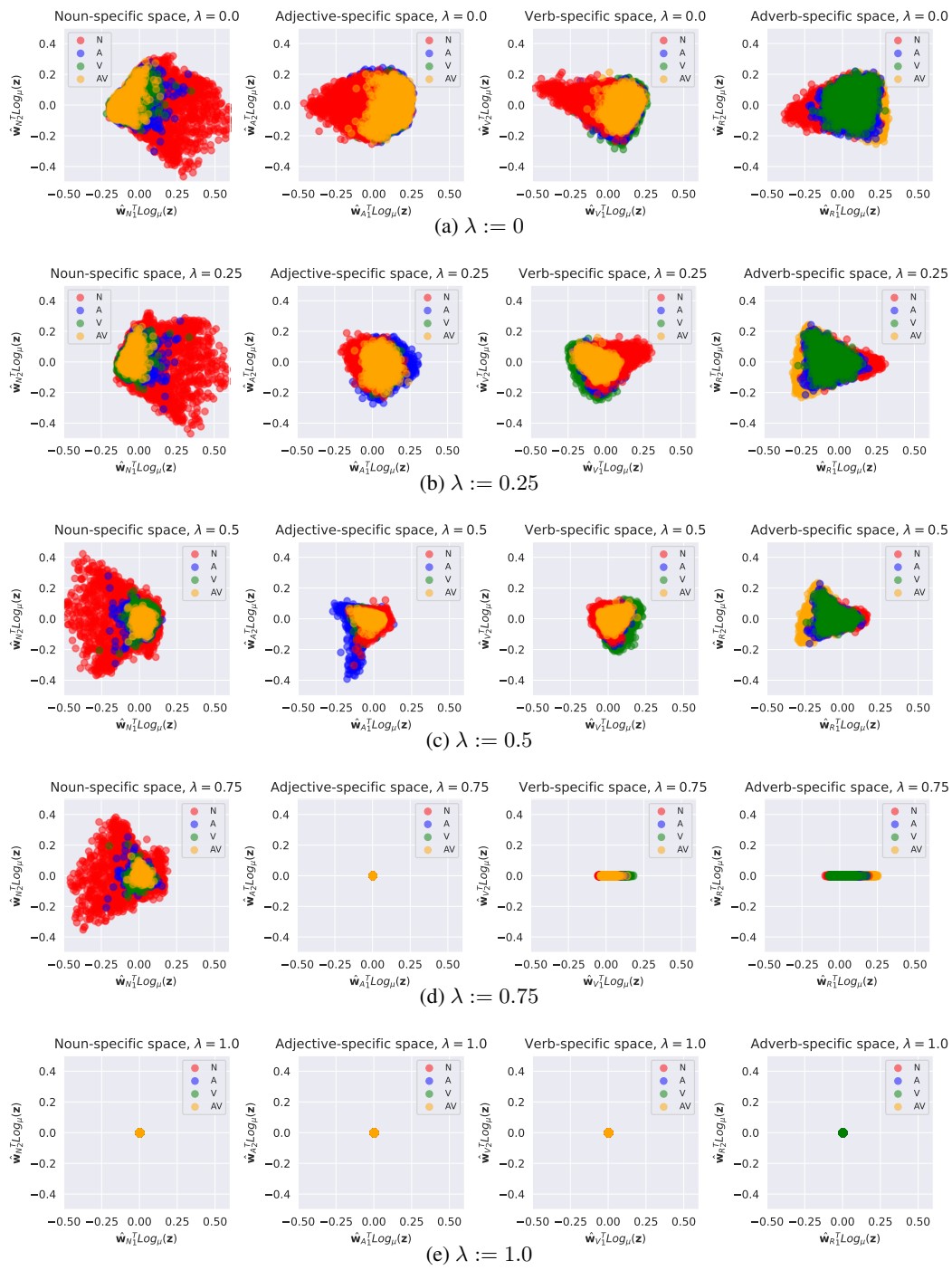

Figure 17: Embeddings' first two coordinates in the tangent space(es), with various values of $\lambda$ in the main objective (axis limits are fixed to compare length of vectors across values of $\lambda$). The base CLIP model `clip-vit-base-patch32` is used here.

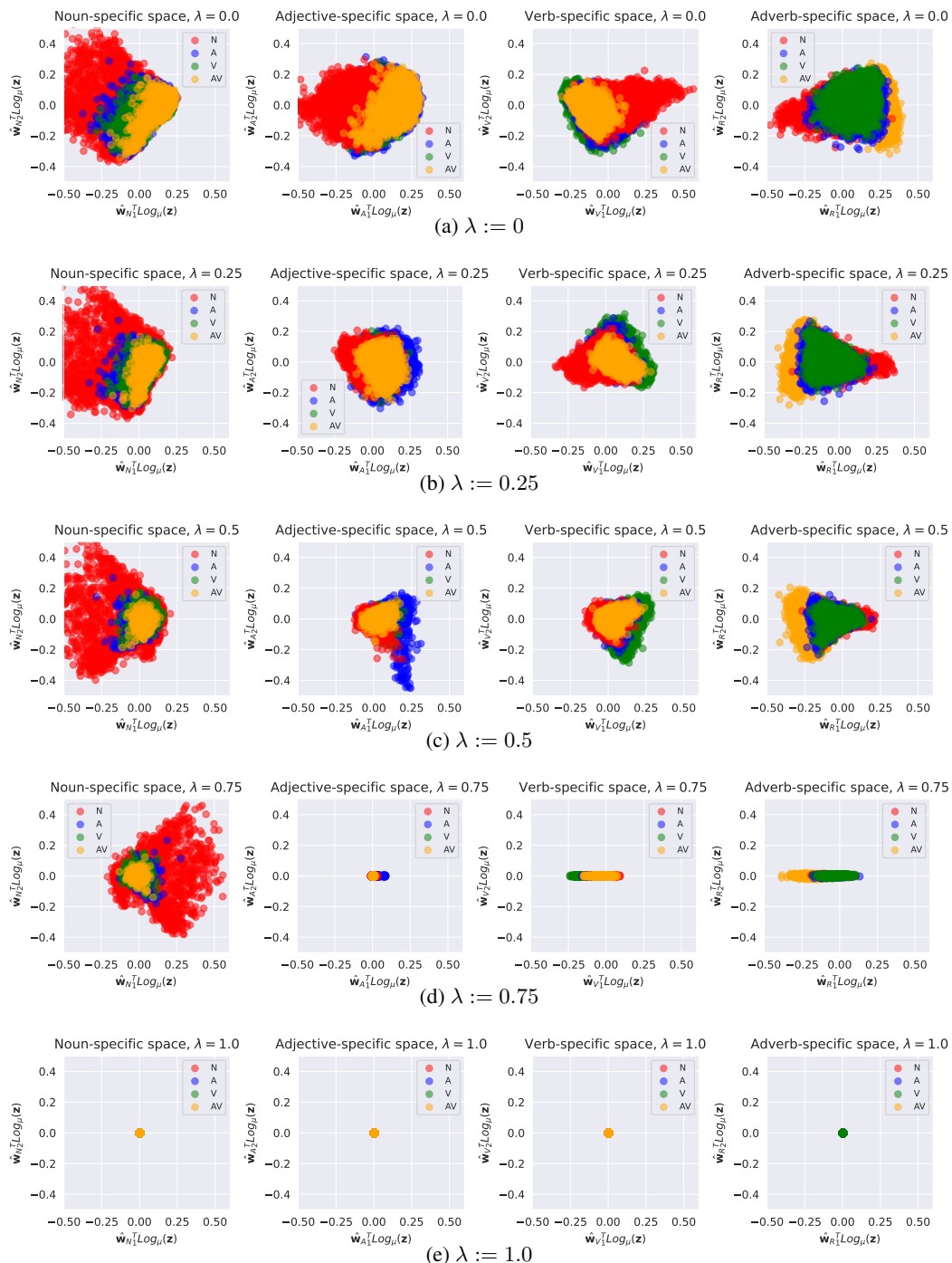

Figure 18: Embeddings' first two coordinates in the tangent space(es), with various values of $\lambda$ in the main objective (axis limits are fixed to compare length of vectors across values of $\lambda$). The larger CLIP model `clip-vit-large-patch14` is used here.

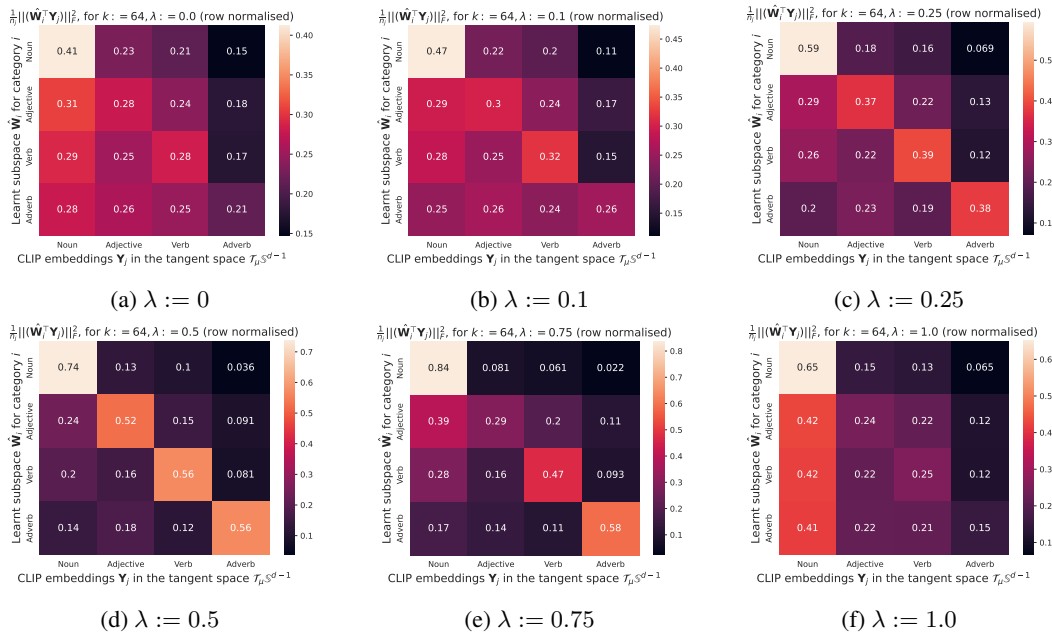

Figure 19: Ablation on $\lambda$ with the quantity $\frac{1}{n_j}||(\hat{\mathbf{W}}_i^\top \mathbf{Y}_j)||_F^2$ introduced in the main paper. Row-normalisation is performed to highlight the relative representation of each class' embeddings within each subspace. The base CLIP model `clip-vit-base-patch32` is used here.

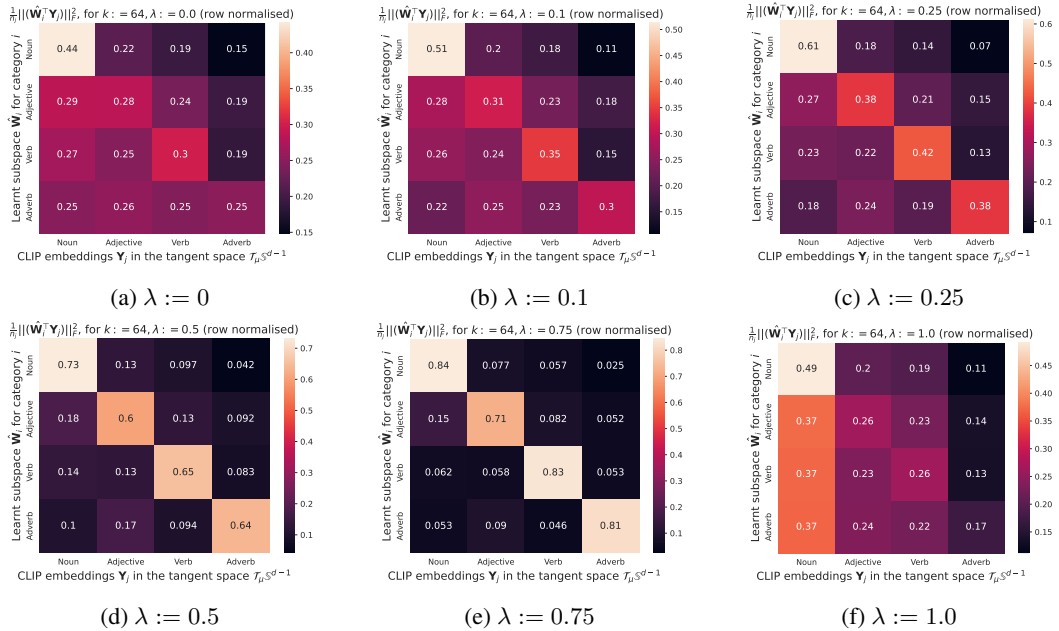

Figure 20: Ablation on $\lambda$ with the quantity $\frac{1}{n_j}||(\hat{\mathbf{W}}_i^\top \mathbf{Y}_j)||_F^2$ introduced in the main paper. Row-normalisation is performed to highlight the relative representation of each class' embeddings within each subspace. The larger CLIP model `clip-vit-large-patch14` is used here.

# D  Experimental details

For reference in this section, we first include the dimensionalities of the shared VL spaces and public links to implementations used of the three CLIP models in this paper in Table 4.

Table 4: Dimensionality of the VL space representations in the four CLIP models.

| Model name | `clip-vit-base-patch32` | `clip-vit-base-patch16` | `clip-vit-large-patch14` | 'OpenCLIP' |
|---|---|---|---|---|
| Dimensionality of $\mathbf{z}_I$, $\mathbf{z}_T$ | 512 | 512 | 768 | 1024 |
| Public link | HuggingFace | HuggingFace | HuggingFace | Github |

**Visual disentanglement**   The Paella TTIM [15] used in the main paper adopts the 'OpenCLIP' [50] model with a larger $d = 1024$-dimensional VL representation. For all 'visual disentanglement' results throughout both the paper and supplementary material, we use $k = 768$ dimensional 'adjective' and 'noun' subspaces for all text prompts (apart from when removing the 'content' representations in visually polysemous phrases, where we find only $k = 32$ components are necessary).

**Visual theme subspaces**   For the custom visual subspaces, we produce a list of phrases related to the visual theme of interest by asking ChatGPT [51] questions of the format: `Please give me a list of 250 words and phrases related to the concept of {x}`, where x is the visual concept of interest (such as 'gore'). For the case of the artist subspace, we ask: `Please give me a list of 250 of the most famous painters and visual artists of all time`. In each scenario, we follow up twice more asking for additional responses (given the limited response length), specifying that it tries not to repeat any of the previous answers in the list. For the experiments in the 'gory' and 'artist' custom subspaces, ChatGPT gave us 371 and 830 unique phrases respectively (taking just the provided artists' surnames as additional examples for the latter), and use a $k = 128$- and $k = 512$-dimensional subspace respectively (given the limited number of phrases for the gore subspace provided by ChatGPT).

**Concurrent work**   Recent preprints [52, 53] explicitly address the task of ablating particular concepts in diffusion models specifically. However, in contrast to the proposed method, these concurrent works fine-tune Stable Diffusion–specific [4] submodules, and do not focus on the final CLIP vector representations. Thus, there is no straightforward way to compare the proposed method working in CLIP's shared vision-language space directly nor the alternative Paella [15] TIIM. One methodological benefit to [52] over the proposed method however (purely in the context of text-to-image synthesis) is in the requirement of only a single text prompt describing a concept, relative to our necessary collection[4]. On the other hand, our subspaces are learnt in closed form–for example, the 'gory' subspace takes only 0.28 seconds to compute on a V100 GPU, given the CLIP embeddings. This is in contrast to [52]'s models which are stated to require 1000 gradient descent steps to compute, and [53] taking 5 minutes per concept.

**Compute time and hardware**   To run the Paella model, we use a 32GB NVIDIA Tesla V100 GPU. Learning the subspaces is particularly fast given the closed-form solution, taking just 1.1 seconds to compute all 4 (noun, adjective, verb, and adverb) PoS subspaces. Encoding all WordNet PoS examples with CLIP takes 28.91 minutes, however, this is a fixed cost and only needs to be done once at the beginning (after which any number of additional subspaces can be computed very quickly).

---

[4]In particular, $n$ phrases' embeddings can span a subspace with a maximum of $n$ dimensions

