# OpenReview forum: "Parts of Speech–Grounded Subspaces in Vision-Language Models"
_NeurIPS.cc/2023/Conference — NeurIPS 2023 poster_

### Official Review · Reviewer_yh2S · 2023-07-01

**Soundness:** 4 excellent
**Presentation:** 4 excellent
**Contribution:** 3 good
**Rating:** 7
**Confidence:** 5

**Summary:**

The paper presents an innovative solution to address the problem of polysemy within CLIP's embedding space. The authors propose a novel approach that involves decomposing CLIP embeddings into distinct subspaces, with each subspace representing a specific part of speech. This decomposition technique enables the isolation of different parts of speech within a given sentence, thereby facilitating subsequent manipulation in downstream tasks. The experimental results presented in the paper demonstrate the efficacy of the proposed approach in effectively eliminating properties associated with particular parts of speech during CLIP's text-to-image generation process.

**Strengths:**

The paper presents a novel approach to decompose the embedding space of CLIP. Theoretical analysis and experimental results provide compelling evidence that the proposed approach effectively disentangles properties related to different parts of speech within the embedding space. This work is of significant value to researchers, as comprehending and manipulating the embedding space learned by deep neural networks is both crucial and challenging. Understanding the features that embeddings can represent and learning how to manipulate them is essential to the improvement of DNNs. The paper is written clearly and is well-structured.

**Weaknesses:**

While the paper's exploration of subspace decomposition focuses on addressing the polysemy issues associated with part of speech in CLIP embeddings, it is important to note that there are instances of polysemy that cannot be disambiguated solely based on part of speech. For example, consider the word "crane," which can refer to both a bird and a machine. These instances present case-by-case ambiguities, and it remains unclear whether the proposed method can be extended to tackle such scenarios successfully as there are no universal subspaces that can disentangle all of them.

**Questions:**

In the context of addressing polysemy and disambiguation, would it be more straightforward to incorporate more detailed descriptions in the prompts? Could you please elaborate on the advantages of the proposed method over using prompts with additional details?

**Limitations:**

Yes

---

> ### Author Rebuttal · Authors · 2023-08-09
>
> We thank the reviewer for the praise of the paper! We address all comments and questions raised in the two sections below:
>
>
> - `it is important to note that there are instances of polysemy that cannot be disambiguated solely based on part of speech. […] it remains unclear whether the proposed method can be extended to tackle such scenarios successfully as there are no universal subspaces that can disentangle all of them.`
>
>
>     * Please see **Fig. 1 of the new rebuttal PDF**, where we show our method can indeed be extended using the idea proposed in the paper of more specific visual subspaces to handle the reviewer’s example of a polysemous noun.
>
>
>         In particular, we show one can learn a custom “animal” subspace following the same protocol in Sect. 3.1.2 of the main paper. When projecting CLIP representations of “a photo of a crane” onto this subspace, we produce animals of the *bird*, rather than of machinery. Conversely, projecting the CLIP representation of “A photo of a bass” onto this subspace’s orthogonal complement *removes* the representation of “bass” as a fish, synthesizing instead just images of bass *guitars*, following the examples in [1]. We hope this provides further indication of the proposed method and the main objective’s value.
>
>         We thank the reviewer for making the insightful comment that the parts of speech alone cannot disambiguate between certain instances of polysemy—we will address this in the paper’s limitations sections and include these new results clarifying the further potential of the custom visual subspaces.
>
>
>
> - `In the context of addressing polysemy and disambiguation, would it be more straightforward to incorporate more detailed descriptions in the prompts? Could you please elaborate on the advantages of the proposed method over using prompts with additional details?`
>
>
>     * Whilst such manual prompt engineering requires additional effort relative to the automated subspace projections, and domain-specific understanding of each dataset (in the case of zero-shot learning for example), there are also more fundamental advantages to working on the CLIP representations directly.
>
>         In particular, the ability to simply manually add additional details to the input prompts is not possible for tasks where it is a *user* with unconstrained natural language control over the input prompt. In such scenarios, it’s therefore necessary to perform the filtering/disambiguation on a later representation level such as with the proposed method. One example of this is the task of blocking stylistic imitation of Sect. 3.1.2, or when preventing the synthesis of e.g. gory imagery.
>
>         Additionally, we highlight that we learn subspaces on the joint vision-language embeddings, allowing the method to also be applied to *image* representations--which do not permit the option to add additional text details to disambiguate/add desired context (for example, it’s the image representations that are projected onto the subspaces in the ZS experiments in section 3.2.2). This means the subspaces would also be applicable to tasks where the sole input is an image, such as CLIP-based image retrieval (e.g. to search based on either style or content similarity only, or to block the matching on NSFW imagery).
>
>     ---
>
>     **[1]:** Chefer, Hila et al. “The Hidden Language of Diffusion Models.” *ArXiv* abs/2306.00966 (2023).

---

> > ### Comment · Reviewer_yh2S · 2023-08-14
> >
> > The authors' responses have addressed my questions. I've updated my rating to accept.

---

### Official Review · Reviewer_Qiru · 2023-07-02

**Soundness:** 2 fair
**Presentation:** 2 fair
**Contribution:** 2 fair
**Rating:** 5
**Confidence:** 2

**Summary:**

The paper gives a closed-form solution to project CLIP representation of image/text into a subspace with disentangled modes. The proposed method is demonstrated qualitatively in text-to-image generation, and quantitatively by zero-shot classification.

**Strengths:**

- The subspace projection proposed by the paper with a closed-form solution can directly be applied to models based on CLIP, without further training.
- Qualitative results demonstrate the effectiveness in some extent.

**Weaknesses:**

- The motivation of this work is CLIP's representation is biased and unpredictable. The paper proposes to learn sub-space representations for content and appearance. There are many diffusion-based works on guided generation for controlling contents and appearances. However, the paper (claims to have wide applications in generation tasks) fails to compare to any current work on this topic.
- Limited quantitative studies, except for zero-shot classification.

In summary, the close-form projection proposed in this paper is simple/fast and (qualitatively) effective in the some examples shown. But the paper lacks comparisons to recent works on diffusion generation with controllable contents and appearances.

**Questions:**

N/A

**Limitations:**

The limitation is discussed, but only for the dimensionality of sub-space (a hyper-parameter k), which is superficial.

---

> ### Author Rebuttal · Authors · 2023-08-09
>
> We thank the reviewer for their praise of the method as simple, fast, and effective. We address the two weaknesses raised below (where [LX] refers to line number X of the submitted paper):
>
>
> - `However, the paper (claims to have wide applications in generation tasks) fails to compare to any current work on this topic. [...] paper lacks comparisons to recent works on diffusion generation with controllable contents and appearances`
>
>
>     * We kindly highlight to the reviewer that **this work is focused on grounding vision-language representations in PoS****, with** **the** **applications mentioned above** **used** **specifically to evaluate said representations.**  Therefore, we respectfully disagree that a comparison to the controllable content/appearance image generation literature is appropriate for evaluating the claims we make in the paper about the PoS subspaces. In more detail, the goal of the paper is to recover the appropriate PoS subspaces that can provide more fine-grained control over the modes of variation in CLIP representations (e.g. [L48]). We *evaluate* this claim qualitatively with TTIMs and quantitatively through class invariance metrics and ZS classification. We expand below on how this pertains to each experiment:
>
>
>         **Visualising the CLIP representations (Fig. 3)**
>
>
>         - As we state throughout the paper (e.g. lines [L18,L190]), the style/content experiments in Fig. 3 serve simply as a **means of validating the learnt CLIP representations** qualitatively. This evaluation protocol follows that of the related works [1, 2] exploring CLIP representations, which both use text-to-image models for the same visualisation purpose but do not compare to image synthesis methods/techniques themselves for the same reasons we outline here.
>
>
>             For example, Reviewer-yh2S highlights that the `experimental results provide compelling evidence that the proposed approach effectively disentangles properties related to different parts of speech within the embedding space`, which is the intention of the qualitative text-to-image experiments in Fig. 3.
>
>
>         **Theme erasing (Figs. 4&5)**
>
>         - The custom theme CLIP subspaces of Sect. 3.1.2 are further evaluated qualitatively in Figs. 4&5 with the theme erasing application. As discussed in the supplementary material, there are two related preprints to first appear on arXiv ~2 months before the submission deadline [3, 4] (we thus consider these “concurrent works” per the NeurIPS guidelines). However, both of these preprints work with specific submodules of the alternative stable diffusion TTIM, therefore there is no straightforward way to adapt the approaches to the CLIP-based Paella model nor to erase concepts from the CLIP feature space for comparison to the proposed method.
>
>
>         Finally, the reviewer quotes [L336] from our “limitations” sections, where we state that: `Whilst the model has wide application for both generative and discriminative tasks, it is not able to perfectly separate the modes of variation for every possible image and text prompt`. We have accordingly changed this to read: `Although the recovered subspaces show wide applicability in downstream tasks, [...]` to better reflect the scope of the paper.
>
> - `Limited quantitative studies, except for zero-shot classification.`
>
>     * We kindly draw the reviewer’s attention to the fact that zero-shot classification is *not* just the only way we evaluate the method quantitatively. A quantitative “class invariance” score is presented for two different applications in both Fig. 6 of the main paper and Fig. 6 of the supplementary material to further demonstrate that the subspaces capture variation in just the specific categories of words of interest.
>
> ----------
>
> **[1]:** Materzynska, Joanna et al. “Disentangling visual and written concepts in CLIP.” *(CVPR)* (2022).
>
> **[2]:** Menon, Sachit et al. “Task Bias in Vision-Language Models.” *ArXiv* abs/2212.04412 (2022).
>
> **[3]:** Gandikota, Rohit et al. “Erasing Concepts from Diffusion Models.” *ArXiv* abs/2303.07345 (Mar 2023).
>
> **[4]**: Kumari, Nupur et al. “Ablating Concepts in Text-to-Image Diffusion Models.” *ArXiv* abs/2303.13516 (Mar 2023).

---

> > ### Comment · Reviewer_Qiru · 2023-08-15
> >
> > Thanks for the response. I've now updated my score accordingly.

---

### Official Review · Reviewer_QWgZ · 2023-07-05

**Soundness:** 4 excellent
**Presentation:** 4 excellent
**Contribution:** 4 excellent
**Rating:** 7
**Confidence:** 4

**Summary:**

The following work proposes a geometry-aware approach to identifying subspace projections within the CLIP embedding space. The projections allow one to limit the CLIP embeddings to the subspaces corresponding to individual parts of speech (noun, adj... ). This allows for more fine-grained controllability when using CLIP embeddings for downstream tasks such as text to image synthesis. Notably, the authors take into account the non-euclidean nature of CLIP embeddings (located on the surface of a hypersphere) by mapping it a specific tangent space first. Experiments demonstrate additional controllability with regards to visual style when given access to a part-of-speech partitioning of the CLIP embedding space.



**Strengths:**

- Principled approach to handling the non-euclidean nature of the CLIP embedding space. Personally I think this is an important topic that is often overlooked in many applied works building on top of CLIP and the recent improved VQGAN architecture, both of which employ a normalized embedding space. While the geometry-aware formulation is perhaps not the most novel contribution of this work, I believe it may serve as an important blueprint for future works relying on normalized embeddings.
- Overall, the closed-form linear formulation of the subspace solving objective provides a low complexity but effective solution to the problem statement.

**Weaknesses:**

- As much as the geometry aware formulation is mathematically justified, it would be nice to see some sort of experiment that demonstrates a significant loss in downstream task performance or even a simple plot based analysis as in the supplementary if one were to ignore it.
- The visualization for lambda selection in supplementary figure 11 is unclear. The plotting software clearly overlays each point cloud on top of the others. As such, it is difficult to visually confirm the spread of all the point clouds except for the last one rendered, which I believe is the yellow adverb cloud. In order to properly visualize this, I think we would have to have juxtaposed separate plots for each cloud with the same axis scaling and offset.
- While I am aware that there are many standards for placement of related works, I think it would be best for an earlier placement before methods given the proposed formulation's close relationship to fisher discriminant analysis.

**Questions:**

See weaknesses.

**Limitations:**

Adequately addressed.

---

> ### Author Rebuttal · Authors · 2023-08-09
>
> We thank the reviewer for their praise of the paper and thorough review. We address the 3 stated weaknesses below:
>
>
> - `it would be nice to see some sort of experiment that demonstrates a significant loss in downstream task performance or even a simple plot based analysis as in the supplementary if one were to ignore it.`
>
>
>     * We kindly draw the reviewer’s attention to Fig. 8 in the supplementary material where we show that the submanifolds lead to higher downstream zero-shot accuracy than the Euclidean subspaces across almost all datasets considered. We highlight that this leads to gains of e.g. ~30% on flowers102 and almost 20% on CIFAR100 (which are non-trivial in the zero-shot setting).
>
>
>         We will make sure to include these results in the main paper with the extra page in the camera-ready version to emphasise these results.
>
> - `In order to properly visualize this, I think we would have to have juxtaposed separate plots for each cloud with the same axis scaling and offset.`
>
>
>     * We thank the reviewer for the helpful suggestion to separate the PoS point clouds within each subplot of Fig. 11 of the supplementary material to better visualise the relative spread—which would indeed be more intuitive.
>
>
>         Our only concern with this suggestion is that this would either lead to Figs. 11&12 spanning 8 pages (if we were to create 4 subplots for each current subplot) or make the figure much more cluttered and thus difficult for the reader to easily compare across values of lambda (if we instead subdivided each existing subplot into quadrants). We will endeavour to incorporate this latter subplot division suggestion and/or explore additional ways of better visualising this for the camera-ready, such as using an opacity < 1.0 for the existing plots to allow one to better view the other PoS point clouds underneath that of the adverb.
>
> - `I think it would be best for an earlier placement before methods given the proposed formulation's close relationship to fisher discriminant analysis.`
>
>
>     * We agree with the reviewer that, all else equal, placing the related work much earlier would be preferred. The reason we eventually opted to place the related work after the methodology section is to be able to contrast the proposed objective more precisely with the related existing methods, having previously introduced the paper’s notation and proposed mathematical formulation. For example, in this position, we can discuss exactly how each $\mathbf{X}_i$ is manipulated by the different methods (the reader better understanding at this point what each matrix refers to in the context of the paper), and reference the equations describing the exact proposed formulation. We found this to be too confusing when the section was placed before the methodology.
>
>
>         Given the close relationship of the component analyses, however, we will add an additional brief, more high-level discussion about the connections to the related work in the introduction, given the extra page allowed in any final version of the paper—we thank the reviewer for the helpful suggestion.

---

> > ### Comment · Reviewer_QWgZ · 2023-08-12
> > **Concerns appropriately addressed**
> >
> > I have read the authors' responses to all reviews and am satisfied with their responses. As such, I retain my original rating of "Accept".

---

### Official Review · Reviewer_3FWH · 2023-07-07

**Soundness:** 3 good
**Presentation:** 3 good
**Contribution:** 3 good
**Rating:** 7
**Confidence:** 5

**Summary:**

This work proposes to learn subspaces for disentangling the visual representation in the CLIP space, based on the parts of speech of the prompt. A closed form solution is presented where the norm corresponding to the embedding of word of interest has a maximum norm while the norm of the rest is minimized. Qualitative results are shown on the  CLIP-based TTIM from LAION where visual results are presented by killing on of the subspaces (noun or adjective) and quantitative through performance on a class invariance metric and through zero-shot classification. The method performs better than the prior art.

**Strengths:**

+ The work is very well written, clearly motivated and well presented.
+ This is the first work which attempts to disentangle the subspaces using POS in CLIP based embedding models.
+ The method is easy to implement with the closed-form solution.
+ Qualitative and quantitative evaluation are performed to show the utility of POS guided subspace projection.


**Weaknesses:**

- Effect of the prompts: From the quantitative results the role of the subspaces is not clear. For example, by removing noun from "Van Gogh" it generates the painting. Painting is also a noun. Therefore, the distinction is not clear.

- In the qualitative examples (Figure 5), the images also change and the semantics on removing one POS subspace do not guarantee that the original image's style is preserved.

- It would be good to show the results with multiple samples from the given prompt on the original dataset to show if the method actually works or just picks up on the partial prompts which would also work with the baseline model.  For example, "A mutlicolored Penguin" and "A penguin" are very general prompts and they can have similar results without the subspace projection.

- "Disentanglement" in multimodal approaches has been presented in prior work [1,2].
1.  Fast, Diverse and Accurate Image Captioning Guided By Part-of-Speech. CVPR 2019.
2. Diverse image captioning with context-object split latent spaces.

**Questions:**

1. How would different partial prompts in the original baseline model compare with the proposed approach as pointed to in weakness?
2. Are there insights into persevering the underlying style of the image generated from the prompt? Example removing only snow from the already generated "snowy" images of NYC?


**Limitations:**

There is no potential negative societal impact of their work.

---

> ### Author Rebuttal · Authors · 2023-08-09
>
> We thank the reviewer for their assessment of the paper as “clearly motivated” and “very well written”. We address below the weaknesses raised and answer the two questions asked:
>
>
> - `Effect of the prompts: From the quantitative results the role of the subspaces is not clear. For example, by removing noun from "Van Gogh" it generates the painting. Painting is also a noun. Therefore, the distinction is not clear.`
>
>     * We respectfully argue that the distinction is clear from our qualitative results. In particular, whilst “painting” is indeed a noun, we do not see any generated “object” instantiation of a painting present *within* the image in Row 2 of Fig. 1, but rather an image with the visual styles associated with the artist. Of course, any non-blank synthetic image will always contain *something* one could call a noun (and ultimately we still see an “image”*,* even though “image” is also a noun), **but crucially this doesn’t refute our claim made in the paper about the specific roles of the subspaces** (e.g. [L199]): that the adjective subspace captures “appearance-based variation” associated with a text prompt, and the noun subspace that of the “objects” described by a text prompt.
>
>         As a concrete example, we see in Row 1 of Fig. 1 that the prompt “Vincent Van Gogh” produces a combination of both the artist themselves and images in their signature style. The former we think of as the “object” associated with the text prompt, and the latter as the visual styles associated with the prompt. We see precisely these two visual components removed when projecting onto the orthogonal complements of the two subspaces in turn. We hope this further clarifies the role of the subspaces.
>
>
> - `In the qualitative examples (Figure 5), the images also change and the semantics on removing one POS subspace do not guarantee that the original image's style is preserved.` **+ Question 2:** `Are there insights into persevering the underlying style of the image generated from the prompt?`
>
>
>     * In Fig. 5, our *goal* is to remove the visual styles associated with the text prompt, and thus the experiments support the claims as intended. We assume the reviewer intended to refer to Fig. 3 (and is asking why we get slightly different images after subspace projection):
>
>         Whilst we don’t claim to address local image editing nor preservation of the original image’s structure, we understand this to be a common phenomenon with TIIMs. For example, [1] states that: `"In particular, even the slightest change in the textual prompt may lead to a completely different output image"` (talking of the SOTA TTIMs). So we view this tendency not as a limitation introduced by the proposed method, but rather one we would inherit if using the method for image editing. If one’s goal was to preserve the image's original structure, incorporating ideas from [1] in freezing a subset of the cross-attention maps would be a sensible approach. We thank the reviewer for pointing this out, and we will add a discussion of this to our limitations section.
>
>         - **[1]:** Hertz, Amir et al. “Prompt-to-Prompt Image Editing with Cross Attention Control.” *ICLR 2023.*
>
>
> - `It would be good to show the results [...] to show if the method actually works or just picks up on the partial prompts which would also work with the baseline model. For example, "A mutlicolored Penguin" and "A penguin" [...] can have similar results without the subspace projection.*` **+ Question 1**
>
>
>     * We politely draw attention to the fact that we already experiment with many examples of text prompts that are *not* decomposable into substrings  (e.g. Figs. 1&3 of the main paper and Fig. 1 of the supplementary material), which support the claim that the method works as intended in separating the latent associations of appearance and content.
>
>
>         As the reviewer correctly points out, however, in some simple descriptive example prompts (such as “a photo of a multicoloured penguin”) it is possible to manually break the prompt down into sub-prompts that describe each target component. We show in **Fig. 2 of the new rebuttal PDF** the outputs when doing this for the requested prompt. Such special cases of prompts can be seen to have a kind of “ground truth”, and thus we suggest that such comparison further validates that our method successfully produces the expected outcome in separating the two visual components.
>
>         It’s important to note however that such **manual prompt engineering does not constitute a replacement to the proposed method**: for example, for the “style-blocking” tasks of Sect. 3.1.2, it is not possible to manually intervene at the text-prompt level given a user has unconstrained natural language control over the input text description. The same holds true of any other task with user-specified text input, or when the input is an image (e.g. in style- or content-based image retrieval).
>
>
> - `Disentanglement" in multimodal approaches has been presented in prior work [1,2].`
>
>     * We thank the reviewer for bringing these two works to our attention. Whilst both works focus on the task of image captioning, which is unrelated and outside the scope of the paper, we will be sure to discuss how both relate to the proposed method in the revised manuscript.
>
>
>         Briefly: [1]’s motivation is to produce “multiple, diverse captions that still properly describe the image”. This is similar in spirit to one problem with CLIP motivating our paper, in that there are multiple equally valid labels associated with an image. However, [1] focus solely on the task of image captioning, proposing a standalone method involving training a new series of networks for this specific task. [2] also proposes a new series of networks for the task of image captioning—attempting instead to disentangle the “object” in an image from the context in which it appears (as opposed to separating style from content, or using PoS).

---

> > ### Comment · Reviewer_3FWH · 2023-08-11
> > **Thank you for the rebuttal**
> >
> > I have read the rebuttal which addresses all the points from the reviewer comments. The paper makes good contributions for controllable generation. I have updated my score accordingly.

---

### Author Rebuttal · Authors · 2023-08-09

We thank all four reviewers for their thorough comments and positive assessment of the paper:


- `Reviewer-3FWH` states that the work is “clearly motivated” and the method “easy to implement”.
- `Reviewer-QWgZ` praises the “principled approach to handling the non-euclidean nature of the CLIP embedding space“ that “may serve as an important blueprint for future work”.
- `Reviewer-Qiru` highlights the simplicity, speed, and effectiveness of the method for some of the qualitative results.
- `Reviewer-yh2S` notes the “work is of significant value to researchers”, and that the experiments and theoretical evidence provide “compelling evidence” of the proposed approach’s ability to disentangle properties relating to the PoS.

In our initial response, we have addressed all weaknesses raised and questions asked by the reviewers, which we hope clarifies any confusion or concerns. We encourage all reviewers to view the **additional 1-page PDF** containing two new figures. **Fig. 1** clarifies the method’s ability to disambiguate between additional instances of polysemy (e.g. “crane”) with the custom subspaces in reply to `Reviewer-yh2S`. **Fig. 2** compares when one synthesises images from sub-prompts describing the individual modes of variation. This further confirms the method works as expected when we have something close to a “ground truth” available for what the disentanglement should look like (as requested by `Reviewer-3FWH`). We are grateful to all reviewers for their time and helpful ideas and we believe the paper will be even stronger after incorporating their comments into any final version.

---

### Decision · Program_Chairs · 2023-09-21

**Decision:**

Accept (poster)

**Comment:**

The paper has received mostly positive scores. The paper aims to disentangle concepts in the latent space of CLIP and experiments are provided especially for part-of-speech disentanglement. The reviewers recognize the usefulness and effectiveness of the method which was also demonstrated through qualitative results through synthesis. The reviewers also appreciate the generality of the proposed method beyond the applications demonstrated in this work. Particularly, preventing a method from copying an artists' style seems in text-to-image models is an important application highlighted by the authors. Overall the paper has a good motivation, methodology and experiments and is recommended for acceptance.